# Learning Initial Basis Selection for Linear Programming via Duality-Inspired Tripartite Graph Representation and Comprehensive Supervision

Anqi Lu [1]   Junchi Yan [1]

## Abstract

For the fundamental linear programming (LP) problems, the simplex method remains popular, which usually requires an appropriate initial basis as a warm start to accelerate the solving process. Predicting an initial basis close to an optimal one can often accelerate the solver, but a closer initial basis does not always result in greater acceleration. To achieve better acceleration, we propose a GNN model based on a tripartite graph representation inspired by LP duality. This approach enables more effective feature extraction for general LP problems and enhances the expressiveness of GNNs. Additionally, we introduce novel loss functions targeting basic variable selection and basis feasibility, along with data preprocessing schemes, to further improve learning capability. In addition to achieving high prediction accuracy, we enhance the quality of the initial basis for practical use. Experimental results show that our approach greatly surpasses the state-of-the-art method in predicting initial basis with greater accuracy and in reducing the number of iterations and solving time of the LP solver.

## 1. Introduction

Linear programming (LP) has played a vital role in decision-making since it was first introduced by (Dantzig, 1948). Many real-world optimization problems can be formulated as linear models or approximated through linearization. Furthermore, LP remains the most frequently used optimization technique in complex optimization problems such as Mixed Integer Programming (MIP).

The simplex method is a widely-used algorithm for LP problems, known for obtaining theoretical optimal solutions represented by basic solutions rather than numerical approximations. With contributions from numerous researchers, the simplex method has maintained high performance on large-scale problems and remains a fundamental approach for solving LP problems. The revised simplex method (Dantzig & Orchard-Hays, 1954) made it practical with limited numerical accuracy on computing units. (Lemke, 1954) demonstrated the importance of the dual simplex method, while (Forrest & Goldfarb, 1992) developed a computationally efficient steepest edge pricing rule. (Harris, 1973) introduced the two-stage ratio test, which uses feasibility tolerances to find better pivot elements. (Suhl & Suhl, 1990) presented an efficient LU factorization implementation, and (Forrest & Tomlin, 1972) improved the updating basis method. Anti-degeneracy techniques (Ryan & Osborne, 1988) prevent the simplex method from cycling in dead loops. These improvements have significantly enhanced the efficiency and stability of the simplex method.

Since the simplex method pivots from basic solution to basic solution, the strategy for choosing the initial basis greatly impacts performance. A better starting point may be closer to the potential optimal solution in terms of logical pivot distance, often resulting in fewer simplex iterations. The most fundamental strategy is the logical basis introduced by (Chvátal, 1983), which simply selects all slack variables. This strategy is straightforward, ensuring nonsingularity and convenient inverse computation, making it still active in simplex solvers like HiGHS (Huangfu & Hall, 2018b). However, this conservative strategy is independent of input problems and does not seek better starting points for various problems. The potential for performance improvement has prompted extensive research.

The seminal work (Bixby, 1992) presented a heuristic method implemented in CPLEX (Gearhart et al., 2013), constructing the initial basis in a preference order. It prioritizes slack and free variables over bounded and non-singleton variables to keep the basis sparse and well-behaved, while minimizing the inclusion of artificial variables to avoid their restrictive bounds on optimization. It works better for easier problems but is less effective for harder ones. The "Idiot" crash (Galabova & Hall, 2020), another practical method, is more complex and time-consuming. Implemented by For-

---

[1]Sch. of Artificial Intelligence & Sch. of Computer Science, Shanghai Jiao Tong University. Correspondence to: Junchi Yan <yanjunchi@sjtu.edu.cn>.

*Proceedings of the $42^{nd}$ International Conference on Machine Learning*, Vancouver, Canada. PMLR 267, 2025. Copyright 2025 by the author(s).

rest in COIN-OR Linear Programming Solver (Clp) (Forrest, 2006), it replaces the linear objective minimization with a combination of the objective and a quadratic function of constraint violations. This method performs better than the primal simplex with a logical basis, especially on quadratic assignment problems (QAPs), but may increase total solving time for other LP problems.

Despite these challenges, researchers continue to focus on improving initial basis selection. In recent years, deep learning—particularly graph neural networks (GNNs)—has shown significant promise across various domains. (Khalil et al., 2017) proposed a GNN model for learning greedy heuristics on several combinatorial optimization problems defined over graphs. (Gasse et al., 2019b) applied GNNs to enhance variable selection policies in branch-and-bound MIP solvers, establishing the use of bipartite graphs of constraints and variables as a standard representation for MIP problems. These efforts highlight the potential of GNNs to learn the structural characteristics of optimization problems. More recently, GNNs have also been applied to initial basis selection for LPs (Fan et al., 2023), demonstrating promising performance compared to traditional heuristics.

However, even when GNNs predict an initial basis close to the optimal one, this does not always lead to greater acceleration. If the predicted basis is invalid, Phase I may require significant time to recover a valid one, which may still be far from optimal. Although predicting a closer basis is an intuitive strategy for acceleration, additional refinements are often necessary for performance improvements.

Our work improves the model's ability to predict a closer basis. More importantly, we go beyond accuracy (closeness) by prioritizing actual solver acceleration. Through a detailed analysis of the LP problem, we introduce additional techniques that significantly enhance practical solving speed, which remains the ultimate objective.

In this study, we aim to further explore the capabilities of GNN-based models in the context of initial basis selection for LP. **Our main contributions are as follows:**

- Inspired by duality in LP, we propose a tripartite graph-based representation for general LP problems. This representation, combined with a specially designed message passing process on graph, enhances the ability of GNNs to represent and aggregate LP problem features.
- We qualitatively analyze the importance of basic variable prediction accuracy on overall prediction accuracy. Additionally, we identify the negative effect of the infeasibility of the initial basis. Building upon these two aspects, we design a new loss function targeted towards basic variable selection and incorporate multi-level labels extracted from the solving path into the loss function.
- We identify potential inconsistencies in raw solver-

derived labels. Based on the analysis, we propose data preprocessing schemes to reduce Bayesian error and improve learning outcomes.
- Beyond higher test prediction accuracy, utilizing the initial basis provided by our model to warm start the state-of-the-art open-source LP solver HiGHS (Huangfu & Hall, 2018a) results in an average over $2\times$ improvement in reducing both average solving iterations and time compared to the SOTA (Fan et al., 2023). Particularly, on the presolved Mirp1 (Papageorgiou et al., 2014) test set, with the solver achieving the optimal solution, our model reduces the average solving iterations to 32% and the average solving time to 42%, outperforming the current SOTA, which achieves 62% and 73%, respectively.

## 2. Preliminaries

We first present the basic for linear programming and its emerging solving techniques leveraging machine learning, though the development is still in its infancy.

### 2.1. Linear Programming

**LP Representation** The general form of LP problems is:

$$\min_{\boldsymbol{x} \in \mathbb{R}^n} \quad \boldsymbol{c}^\top \boldsymbol{x}$$
$$s.t. \quad \boldsymbol{l}^s \le \boldsymbol{A}\boldsymbol{x} \le \boldsymbol{u}^s, \qquad (1)$$
$$\boldsymbol{l}^x \le \boldsymbol{x} \le \boldsymbol{u}^x,$$

where $\boldsymbol{A} \in \mathbb{R}^{m \times n}$, $\boldsymbol{c} \in \mathbb{R}^n$, $\boldsymbol{l}^s$ and $\boldsymbol{l}^x \in (\mathbb{R} \cup \{-\infty\})^n$, $\boldsymbol{u}^s$ and $\boldsymbol{u}^x \in (\mathbb{R} \cup \{\infty\})^n$. We refer to the elements in $\boldsymbol{x}$ as variables and the rows in $\boldsymbol{A}\boldsymbol{x}$ as constraints. There are $n$ variables and $m$ constraints. By defining $\boldsymbol{s} = \boldsymbol{A}\boldsymbol{x}$, we can regard each constraint as a variable in $\boldsymbol{s}$. We can express $\boldsymbol{x}$ as $(\boldsymbol{x}^+ - \boldsymbol{x}^-)$, where $\boldsymbol{x}^+ \ge 0$ and $\boldsymbol{x}^- \ge 0$ represent the positive and negative parts of $\boldsymbol{x}$, respectively.

**Duality** A.1 provides additional details on the duality in LP. The dual problem of the primal LP, expressed in its general form, can be written as

$$\min_{\boldsymbol{y}_l^s, \boldsymbol{y}_u^s, \boldsymbol{y}_l^x, \boldsymbol{y}_u^x \ge 0} - (\boldsymbol{l}^s)^\top \boldsymbol{y}_l^s + (\boldsymbol{u}^s)^\top \boldsymbol{y}_u^s$$
$$- (\boldsymbol{l}^x)^\top \boldsymbol{y}_l^x + (\boldsymbol{u}^x)^\top \boldsymbol{y}_u^x$$
$$s.t. \quad \boldsymbol{A}^\top (\boldsymbol{y}_l^s - \boldsymbol{y}_u^s) + (\boldsymbol{y}_l^x - \boldsymbol{y}_u^x) = \boldsymbol{c}.$$

**Basis** We define $\mathcal{B}_x \subset [n]$ and $\mathcal{B}_s \subset [m]$ to represent the indices of basic variables in variable and constraint sets. Together, $\mathcal{B}_x$ and $\mathcal{B}_s$ form a basis $\mathcal{B} = (\mathcal{B}_x, \mathcal{B}_s)$ for the LP problem, which should satisfy that the number of basic variables equals $m$, i.e., $|\mathcal{B}_x| + |\mathcal{B}_s| = m$, and the basis matrix $\boldsymbol{B} = [\boldsymbol{A}_{\mathcal{B}_x} \; -\boldsymbol{I}_{\mathcal{B}_s}^m]$, which is the concatenation of $\boldsymbol{A}_{\mathcal{B}_x}$ and $-\boldsymbol{I}_{\mathcal{B}_s}^m$, is nonsingular. We define $\mathcal{N}_x = [n] \setminus \mathcal{B}_x$ and $\mathcal{N}_s = [m] \setminus \mathcal{B}_s$ to represent the indices of non-basic variables in their respective sets. Once the basis is

determined, all non-basic variables and constraints are set to their finite upper or lower bounds, and the values of the basic variables can be solved using equation

$$[\boldsymbol{x}_{\mathcal{B}_x}^\top \ \boldsymbol{s}_{\mathcal{B}_s}^\top]^\top = [\boldsymbol{A}_{\mathcal{B}_x} \ -\boldsymbol{I}_{\mathcal{B}_s}^m]^{-1}(\boldsymbol{I}_{\mathcal{N}_s}^m \boldsymbol{s}_{\mathcal{N}_s} - \boldsymbol{A}_{\mathcal{N}_x}\boldsymbol{x}_{\mathcal{N}_x}).$$

If the value of any basic variable lies outside its lower and upper bounds, we call the basis infeasible; otherwise not.

**Simplex Method.** It starts from a feasible basis, and compute the objective value in terms of non-basic variables

$$\begin{aligned}\boldsymbol{c}^\top \boldsymbol{x} =& [\boldsymbol{c}_{\mathcal{B}_x}^\top \ 0]\boldsymbol{B}^{-1}\boldsymbol{I}_{\mathcal{N}_s}^m \boldsymbol{s}_{\mathcal{N}_s} \\ & - ([\boldsymbol{c}_{\mathcal{B}_x}^\top \ 0]\boldsymbol{B}^{-1}\boldsymbol{A}_{\mathcal{N}_x}\boldsymbol{x}_{\mathcal{N}_x} - \boldsymbol{c}_{\mathcal{N}_x}^\top)\boldsymbol{x}_{\mathcal{N}_x}\end{aligned}$$

As long as changing a non-basic variable to a basic variable can reduce the objective value, the simplex method selects one to enter the basis and identifies a basic variable to leave the basis. By doing this, the method transitions from one feasible basis to a neighboring feasible basis. This process iterates until no further reduction in the objective value is possible. This iterative procedure can be viewed as moving from one vertex to another vertex of the feasible region, which forms a simplex.

Note that if the initial basis is infeasible, we need to use simplex phase 1 to convert it into a feasible basis. For more details about the advanced simplex algorithm for general form LP, interested readers can refer to (Ping-Qi, 2014).

### 2.2. GNN-based Model for MIP

Recently, GNN-based models have gained widespread use in the realm of MIP. In MIP problems, variables and constraints interact through the constraint matrix, allowing their interactions to be represented as a graph. (Gasse et al., 2019a) proposed encoding mixed-integer linear optimization problems as a bipartite graph $(\mathcal{G}, C, E, V)$. Each constraint and variable is represented as a node, forming the node sets $C$ and $V$, respectively. Each non-zero value in the constraint matrix creates an edge in the set $E$, where the weight of an edge connecting $C_i$ and $V_j$ is the value of the constraint matrix at row $i$ and column $j$. Lots of works have been done using GNN based on this bipartite representation, including learning to branch (Achterberg et al., 2005) (Gupta et al., 2020) (Nair et al., 2020), learning cut selection (Berthold et al., 2022)(Paulus et al., 2022) (Wang et al., 2023b), learning to presolve (Liu et al.), data generation (Wang et al., 2023a), LP-related problems (Li et al., 2022) (Fan et al., 2023) (Qian et al., 2024) and other MIP problems (Khalil et al., 2022) (Zhang et al., 2023). (Ding et al., 2020) proposed a tripartite graph representation consisting of variables, constraints, and a global node.The edge weights between variables and the global node represent the coefficients of the variables in the objective function, while the edge weights between constraints and the global node represent the right-hand side (RHS) of the constraints.

On the theoretical side, (Chen et al., 2022) demonstrated that GNNs based on bipartite graphs can learn solutions that are infinitesimally close to the $L_2$-norm minimal optimal solutions for LP problems. (Qian et al., 2024) discovered that GNNs can simulate iterations of the interior point method by utilizing message passing on tripartite graphs, yielding solutions closer to optimal LP solutions compared to those obtained from bipartite graphs. This approach also reduces computational costs compared to neural networks employing differential equation systems (Wu & Lisser, 2023).

### 2.3. GNN-based Model for Inital Basis Selection

Here we introduce the state-of-the-art GNN-based model (Fan et al., 2023) for selecting initial bases in LP. In the optimal basis obtained by the solver, each constraint or variable corresponds to an integer value $L(s_i)$ or $L(x_j) \in \{0, 1, 2\}$, where a value of 1 represents a basic variable, totaling $m$, while the remaining non-basic variables take on 0 (or 2), indicating whether they are set to the lower (or upper) bound in the optimal basis. Convert the optimal basis obtained from the solver into a one-hot vector $\boldsymbol{y}_{s_i}$ or $\boldsymbol{y}_{x_j} \in \{0, 1\}^3$ as the label for supervised learning.

Constructing the general LP problem as a bipartite graph, the constraint matrix is transformed into edges, forming the edge set $E$, while other information is converted into node features, forming the sets of variables and constraints $V$ and $C$, respectively. Bidirectional message passing is employed on the bipartite graph. In each layer, information is passed once from variables to constraints and once from constraints to variables for node updates. Similar to the three-classification problem on the graph, each node ultimately outputs a three-dimensional vector, $\boldsymbol{p}(s_i)$ or $\boldsymbol{p}(x_j)$, representing the probabilities of belonging to each of the three categories. The constraints or variables corresponding to the largest $m$ values in $\{\boldsymbol{p}(s_i)[1] \mid 1 \leq i \leq m\} \cup \{\boldsymbol{p}(x_j)[1] \mid 1 \leq j \leq n\}$ are chosen as basic variables. Among the remaining variables, if $\boldsymbol{p}(x_i)[0] > \boldsymbol{p}(x_i)[2]$, then the non-basic variable takes on the lower bound; otherwise, it takes on the upper bound.

A loss function is constructed using the cross-entropy function for the 3-class classification problems:

$$\begin{aligned}\mathcal{L} =& \ l((P), (\boldsymbol{x}, \boldsymbol{s}), (\boldsymbol{y})) \\ =& \sum_{i=1}^m \alpha(\boldsymbol{y}_{s_i})l_{CE}(\boldsymbol{p}(s_i), \boldsymbol{y}_{s_i}) \\ & + \sum_{j=1}^n \alpha(\boldsymbol{y}_{x_j})l_{CE}(\boldsymbol{p}(x_j), \boldsymbol{y}_{x_j}),\end{aligned} \qquad (2)$$

where $\alpha(y_{s_i})$ and $\alpha(y_{x_j})$ represent the weights of different variables, which are inversely proportional to the frequencies of their label categories in the samples.

# 3. Proposed Method

## 3.1. Tripartite Graph Representation of general LP

In the bipartite graph representation used in (Fan et al., 2023), edge information only includes the constraint matrix, whereas crucial data from the objective function coefficients and the bounds of variables and constraints can only be utilized through node features during GNN's message passing. Directly incorporating this information into the graph structure could enhance the representation capability for LP.

The existing tripartite graph representation in (Ding et al., 2020) can only represent LP problems where all constraints are less than or equal to and variable ranges are non-negative. However, in practice, LP problems are often represented in general form, and advanced solvers support direct input of this form. Converting a general form problem to a more standardized form increases the problem's size, inflating the number of variables and constraints. In such cases, even if the neural network can predict good results, the increased problem size input to the solver may negatively impact performance more than the neural network's assistance, resulting in lower overall efficiency.

Based on duality in LP, by comparing the primal problem with the dual problem, it can be observed that each finite upper or lower bound in the primal problem corresponds to a variable in the dual problem. A finite lower bound in the primal problem corresponds to a variable with the coefficient of the negative of this bound in the dual problem's objective function, while a finite upper bound corresponds to the bound itself. Each variable in the primal problem corresponds to an equality constraint in the dual problem, which can also be seen as a constraint with equal upper and lower bounds. Furthermore, combining the two variables corresponding to the constraints' upper and lower bounds in the dual problem forms $(\boldsymbol{y}_l^s - \boldsymbol{y}_u^s)$, whose coefficients in the constraint matrix $\boldsymbol{A}^\top$ match the coefficients of $(\boldsymbol{x}^+ - \boldsymbol{x}^-)$ in the constraint matrix $\boldsymbol{A}$ in the primal problem. Similarly, regarding the bounds of variables as constraints, $\boldsymbol{I}^n$ appears both in the dual and primal problems.

In Fig. 1(a) a tripartite graph representation for general form LP problems is proposed. Each variable in the primal problem corresponds to two nodes in the graph, representing the variable and its upper and lower bounds, forming the node sets $V_{primal}$ and $V_{dual}$, respectively. Each constraint in the primal problem corresponds to one node in the graph, forming the set $C$. Additionally, there is a global node $O$. All nodes can be divided into three parts: $O$, $V_{primal}$, and $V_{dual} \cup C$, with edges connecting nodes between these sets but no edges within each set, forming a tripartite graph.

Similar to the bipartite representation, edges between $V_{primal}$ and $C$ are constructed using the constraint matrix $A$. Each pair of nodes in $V_{primal}$ and $V_{dual}$ representing the

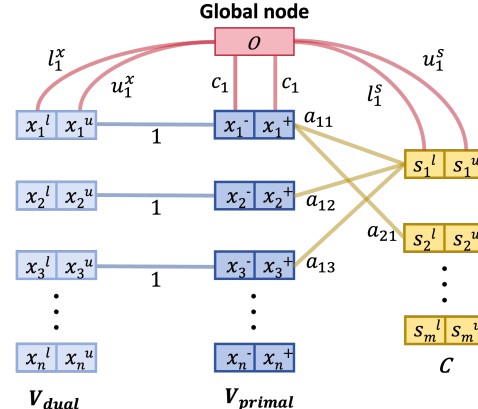

(a) tripartite graph representation of LP

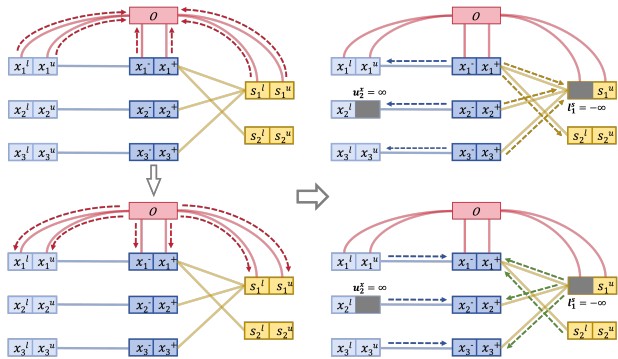

(b) message passing on the graph

*Figure 1.* **Tripartite graph-based GNN:** The objective function and variable bounds are encoded as graph edges, while nodes except the global one are split in half within the tripartite graph, allowing for the representation of both the primal and dual LP problems. A global message passing process begins with the global node, followed by bidirectional message passing between variables and constraints with specific masks based on duality.

same variable are connected by an edge with a weight of 1.

In the connections involving the global node $O$, each node in $V_{dual}$ corresponding to $x_j$ is split into two half-nodes, $x_j^l$ and $x_j^u$, corresponding to $-(y_l^x)_i$ and $(y_u^x)_i$ in the dual problem, connected to $O$ with edges weighted $l_j^x$ and $u_j^x$, respectively. Each node in $C$ corresponding to $s_i$ is split into $s_i^l$ and $s_i^u$, corresponding to $-(y_l^s)_i$ and $(y_u^s)_i$ in the dual problem, connected to $O$ with edges weighted $l_i^s$ and $u_i^s$, respectively. Each node in $V_{primal}$ corresponding to $x_j$ is split into $x_i^-$ and $x_i^+$, representing $-(x^-)_i$ and $(x^+)_i$ in the primal problem, connected to $O$ with edges weighted $c_j$ and $c_j$, respectively. Note that if any of the bounds $l_j^x$, $u_j^x$, $l_i^s$, $u_i^s$ is $\infty$, we do not connect the corresponding edge since there is actually no corresponding variable in the simplified dual problem.

In this tripartite graph representation, the coefficients of the objective function and the bounds of variables and constraints in the primal problem are embedded in the graph through edges connected to the global node. This representation not only captures all information from the original LP problem but also from the dual problem. In the primal problem, $V_{primal}$ and $V_{dual} \cup C$ represent variables and constraints (including variable bounds), respectively. In the dual problem, $V_{primal}$ and $V_{dual} \cup C$ represent constraints and variables. The edges between these sets record the constraint matrices of both the primal and dual problems. The edges between $O$ and $V_{primal}$ reconstruct the objective function of the primal problem and the equality constraints of the dual problem, while the edges between $O$ and $V_{dual} \cup C$ reconstruct all constraints of the primal problem and the objective function of the dual problem.

## 3.2. Message Passing on the Graph

We apply (Qasim et al., 2019) to construct the GNN. We chose this architecture for three main reasons:

**Comparative Consistency:** This architecture is also used in the state-of-the-art work by Fan et al., which our method builds upon. To ensure a fair and direct comparison, we adopted their GNN design as a baseline.

**Analytical Interpretability:** We provide a detailed analysis of the message passing behavior of this architecture in Appendix A.2. In particular, we aim for the amount of message passing to be proportional to the magnitude of the corresponding coefficients in the constraint matrix and the projection of changes in variables (or constraints) onto the respective rows (or columns). This behavior aligns well with the structure and nature of LP problems.

**Computational Efficiency:** Compared to recent models such as TransformerConv (Shi et al., 2020), our architecture is more lightweight and computationally efficient, making it well-suited for practical applications. Furthermore, as shown in Table 14 and discussed in Section A.5, GraphConv achieves slightly better average performance than TransformerConv in our experiments.

In the first step, nodes in $V_{primal}$, $V_{dual}$ and $C$ aggregate information to $O$. Then, information is transmitted from the global node to the nodes in $V_{primal}$, $V_{dual}$ and $C$. With more informative node features, bidirectional message passing between $V_{primal}$ and $V_{dual} \cup C$ is performed. Before each round of bidirectional message passing, masks are applied to the half-nodes in $V_{dual}$ and $C$ that correspond to infinite bounds in the primal problem, ensuring they do not participate in the process. The message passing process is illustrated in Fig. 1(b), with more details provided in A.2.

## 3.3. Loss function for basic variable selection

Different from traditional 3-class classification problems, for each variable or constraint, the category corresponding to the maximum probability output by the corresponding node prediction is not directly selected. Rather, it is ensured that the number of nodes with predicted category 1 exactly equals $m$. In practice, firstly, $m$ basic variables are determined. For the remaining $n - m$ nodes, there is no constraint on the selection between categories 0 and 2. In the process of selecting the $m$ basic variables, if an actual non-basic variable is chosen, this prediction error will not only affect the current variable, but also result in an actual basic variable being predicted as a non-basic variable. So the prediction of basic variables has a greater impact on overall accuracy than the prediction of non-basic variables.

Based on the prediction results, all nodes in $V_{primal}$ are divided into four disjoint sets: $V_{bb}$, $V_{bn}$, $V_{nn}$, and $V_{nb}$, representing nodes where the true category is 1 and the predicted category is 1, where the true category is 1 but the predicted category is not 1, where the true category is not 1 and the predicted category is not 1, and where the true category is not 1 but the predicted category is 1. Similarly, all nodes in the $C$ set are classified into four categories, forming $C_{bb}$, $C_{bn}$, $C_{nn}$, and $C_{nb}$.

In accordance with the deviation between the prediction results and the labels, we aim for the nodes in $V_{bn}$ and $C_{bn}$ to increase the probability of being predicted as category 1, thereby increasing the likelihood of being selected as a basic variable. Conversely, we want the nodes in $V_{nb}$ and $C_{nb}$ to decrease the probability of being predicted as category 1, making it less likely to be chosen as a basic variable. Based on this, we design a loss function component for basic variable selection.

$$
\begin{aligned}
\mathcal{L}_{bas} = {} & \frac{1}{4m_{bV}} \sum_{x \in V_{bn}} l_{BCE}(\boldsymbol{p}(x)[1], 1) \\
& + \frac{1}{4m_{bC}} \sum_{s \in C_{bn}} l_{BCE}(\boldsymbol{p}(s)[1], 1) \\
& + \frac{1}{4(n - m_{bV})} \sum_{x \in V_{nb}} l_{BCE}(\boldsymbol{p}(x)[1], 0) \\
& + \frac{1}{4(m - m_{bC})} \sum_{s \in C_{nb}} l_{BCE}(\boldsymbol{p}(s)[1], 0),
\end{aligned}
\tag{3}
$$

where $m_{bV}$ and $m_{bC}$ represent the number of actual basic variables in the variables and constraints, respectively, satisfying the condition $m_{bV} + m_{bC} = m$.

## 3.4. Loss function for feasibility

The simplex method iterates between feasible bases to approach the optimal solution. Besides obtaining the optimal basis, each vertex on the solving path corresponds to a feasi-

ble basis. These bases sampled along the solution path can be used as additional labels to compute a new loss function term that increases the similarity between the predicted basis and these feasible bases, thereby enhancing its feasibility.

Assuming the total number of iterations in the solver's solving process for the current LP problem is $T$, a total of $N$ feasible solutions are uniformly sampled between iteration $\lceil \theta T \rceil$ and $T$, where $\theta \in (0, 1)$ represents the threshold ratio of starting sampling points to the total number of iterations. For convenience, evenly spaced iterations can be sampled back from the optimal solution to determine the sampling rounds needed. The step size can be set as $\Delta T = \lfloor (1 - \theta)T/N \rfloor$, where the feasible bases obtained at the $T - k\Delta T$ iteration steps ($k \in [N-1]$) are sampled, converted into level $k$ labels denoted as $\boldsymbol{y}_k$. The loss function incorporates these labels and is calculated as

$$\mathcal{L}_{multi} = \sum_{k=0}^{N-1} \mu_k l((P), (\boldsymbol{x}, \boldsymbol{s}), (\boldsymbol{y}_k)), \qquad (4)$$

where, $\mu_k$ represents the weight of the loss calculated from different labels, which should decrease as $k$ increases to ensure that feasible bases closer to the optimal basis have a greater impact on the loss.

By this loss, the network not only learns how to approximate the optimal basis but also learns the path through which the optimal basis is approached. The differences in prediction accuracy corresponding to different basis qualities can vary significantly. For instance, a feasible basis located in the middle of the solving path may have around 50% difference from the final optimal basis, yet using it as an initial basis for warm start can reduce the number of iterations by half. However, an initial basis that is very close to the optimal basis might incur a significant cost in phase 1, potentially leading to a feasible basis that deviates far from the optimal basis. This can result in a sharp increase in computational complexity, leading to a total number of solver iterations exceeding the default mode. In the original loss function, the quality of the prediction results depends solely on the deviation from the optimal basis, without distinguishing between initial bases with the same deviation but different actual qualities. The new loss function term strengthens the labels values that appear multiple times in these labels by summing the weights, guiding the neural network to output the same prediction values. The repeated occurrence of label values signifies their stability along the solving path, implying that opting for these values in the initial basis can reduce infeasibility and facilitate the solver to go back to the solving path with fewer iterations in phase 1.

### 3.5. Data Preprocessing

To some extent, predicting the optimal basis in LP problems is more challenging than predicting the optimal solution. A detailed analysis of this prediction problem and the labels provided by LP solvers is presented in A.3. Inconsistency in labels arising from the structure of the problem is commonly found in practical LP problems. It stems from the fact that the representation of LP problems is not compact enough and contains redundant information. It is worth noting that advanced solvers often preprocess LP problems by eliminating redundant information, leading to a more concise problem representation. This initial time investment typically results in significant time savings during the subsequent solving process. Therefore, utilizing the LP problem obtained after presolving as the dataset not only enhances the learning efficacy of neural networks but also aligns with the standard solving process in practical applications.

Label inconsistency also arises when the upper and lower bounds of variables or constraints are equal. However, when the upper and lower bounds of constraints are equal, representing an equality constraint, these constraints cannot be removed during presolve. Therefore, additional processing is required to address the label inconsistency corresponding to the constraints.

Identifying all equality constraints based on the upper and lower bounds of constraints in the LP problem and excluding the ones that act as basic variables in the optimal basis result in a set of constraints needing preprocessing, denoted as $C_{preprocess}$. Since these constraints have equal upper and lower bounds, solutions corresponding to taking the upper and lower bounds will be identical, leading to equivalent bases. To ensure label consistency, their labels are set to 0.

By labeling the equality constraints in this manner, the possible label values for constraints with equal upper and lower bounds can only be 0 or 1. Consequently, before obtaining the probabilities of the three categories through softmax for the three-dimensional logits output by the neural network during inference, an additional knowledge-based mask can be incorporated according to the input as

$$\text{Mask}_{s_i} = \begin{cases} [0, 0, -\infty]^\top, & \text{if } l_i^s = u_i^s \\ [0, 0, 0]^\top, & \text{otherwise.} \end{cases}$$

**Comprehensive supervised training process:** With a loss function for basic variable selection, another for feasibility, and the label preprocessing, each of which is tailored to the prediction, warm start, and training tasks respectively, we can train the GNN models following the process in Fig. 2.

## 4. Experimental Results

### 4.1. Experimental Setup

**LP Solver:** Access to commercial LP solvers like Gurobi (Gurobi Optimization, LLC, 2023) and CPLEX (Cplex, 2009) is often restricted in some cases e.g. for detailed hyperparamter setting etc. Therefore, in line with the peer

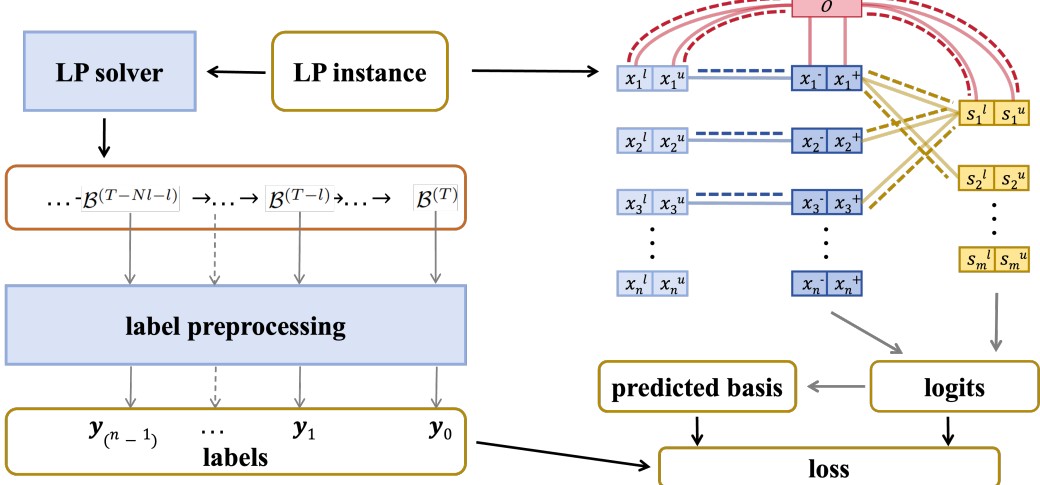

*Figure 2.* **Illustration of the comprehensive supervised training process:** The LP instance is encoded as a tripartite graph and solved by the LP solver. Feasible bases are selected along the solving path to create preprocessed multiple labels. The 3-dimensional logits generated by the GNN, the predicted basis derived from the logits, and the multi-labels are collectively used to calculate the loss function.

work (Fan et al., 2023), we utilize **HiGHS** (Huangfu & Hall, 2018a), a state-of-the-art open-source solver specialized in efficient solving of LP problems. We employ the dual simplex method, as it is typically much more efficient than the primal simplex method.

**datasets:** We utilize well-known publicly available MIP datasets in our study, including the **Load Balance**, **Anonymous**, and **Maritime Inventory Routing Problem** (MIRP) (Papageorgiou et al., 2014) (Jiang & Grossmann, 2015). The **Load Balance** and **Anonymous** datasets are sourced from the NeurIPS 2021 Competition (Gasse et al., 2022). Due to its limited scale with solving iterations in the relaxed LP consistently below 1000, we exclude the **Item Placement** dataset from these competition datasets. We specifically choose the validation dataset of **Load Balance** due to the high similarity among its instances, eliminating the necessity for a larger training dataset. The MIRP datasets consist of three sets of instances labeled as Mirp1 (Papageorgiou et al., 2014), Mirp2 (Papageorgiou et al., 2014), and Mirp3 (Jiang & Grossmann, 2015). We opt for the first two sets as most of the instances in the third set are significantly smaller in scale. We only utilize the Mirp1 dataset from the most closely related ML method (Fan et al., 2023), as it is the only one publicly available in that work.

We relax these MIP problems to obtain the original LP datasets. Additionally, we use HiGHS to presolve the relaxed LP instances, as presolving is commonly applied before solving in practical scenarios. Our datasets include both the original unpresolved instances and the presolved ones. To filter out very small instances, we exclude those with fewer than 2000 solving iterations. Using the same method

in (Fan et al., 2023), we split each dataset into training and test sets in a 7:3 ratio. Additional details about the datasets are available in A.4.

**Compared Methods:** There are few existing works for learning to select initial basis, except for (Fan et al., 2023) which is based on a **bipartite graph-based GNN**. This method has already demonstrated better performance compared to traditional heuristic-based methods in many cases. In our experiments, we compare our **tripartite graph-based GNN** model with this state-of-the-art (SOTA) model. **B**, **M**, and **P** respectively represent the incorporation of the **loss function for basic variable selection**, the **loss function for feasibility**, and **label preprocessing**. For consistency in comparison, we adopt the training hyperparameters outlined in the SOTA method (Fan et al., 2023) for all models. The value of $\mu_k$, defined in Equation 16, is selected based on empirical experience rather than precise fine-tuning. Further details about the hyperparameters and GNN architectures can be found in Appendix A.5.

**Implementation**: The GNN models are trained on NV A100 GPU. The solver operates on a system equipped with a 20-core CPU (Intel Xeon @ 2.90GHz) and 128GB of RAM. Our code is publicly available[1].

### 4.2. Model Evaluation

We compare the bipartite graph-based GNN model using default training with our tripartite graph-based GNN model using comprehensive supervision. Table 1 and Table 2 present the test performance of both models on the unpresolved and

---

[1]https://github.com/HAHHHD/TripartiteLP

*Table 1.* Test performance of the GNN prediction model on the unpresolved datasets.

| | bipartite | | | | tripartiteBMP | | | |
|---|---|---|---|---|---|---|---|---|
| | Precision | Recall | Acc | BasAcc | Precision | Recall | Acc | BasAcc |
| Mirp1 | 0.915 | 0.902 | 0.902 | 0.914 | 0.931 | 0.935 | 0.939 | 0.937 |
| Mirp2 | 0.788 | 0.791 | 0.812 | 0.649 | 0.782 | 0.786 | 0.876 | 0.639 |
| Anonymous | 0.852 | 0.858 | 0.841 | 0.808 | 0.863 | 0.872 | 0.895 | 0.821 |
| Load Balance | 0.932 | 0.979 | 0.989 | 0.989 | 0.910 | 0.992 | 0.986 | 0.986 |
| Geomean | **0.870** | 0.880 | 0.883 | 0.830 | **0.870** | 0.893 | 0.923 | 0.834 |

*Table 2.* Test performance of the GNN prediction model on the presolved datasets.

| | bipartite | | | | tripartiteBMP | | | |
|---|---|---|---|---|---|---|---|---|
| | Precision | Recall | Acc | BasAcc | Precision | Recall | Acc | BasAcc |
| Mirp1 | 0.909 | 0.894 | 0.914 | 0.910 | 0.923 | 0.928 | 0.935 | 0.930 |
| Mirp2 | 0.810 | 0.812 | 0.876 | 0.690 | 0.812 | 0.814 | 0.883 | 0.695 |
| Anonymous | 0.859 | 0.848 | 0.872 | 0.801 | 0.857 | 0.859 | 0.893 | 0.794 |
| Load Balance | 0.945 | 0.942 | 0.924 | 0.943 | 0.946 | 0.942 | 0.925 | 0.943 |
| Geomean | 0.879 | 0.873 | 0.896 | 0.830 | **0.883** | **0.884** | **0.909** | **0.834** |

*Table 3.* Number of iterations of the solver with/without warm start using the predicted initial basis on the unpresolved datasets.

| | default | bipartite | | | tripartiteBMP | | |
|---|---|---|---|---|---|---|---|
| | Iters | Iters | Ratio | Improv | Iters | Ratio | Improv |
| Mirp1 | 25432 | 13711 | 53.91% | 85.49% | **10584** | 41.62% | 140.3% |
| Mirp2 | 86897 | 85691 | 98.61% | 1.41% | **81010** | 93.23% | 7.27% |
| Anonymous | 35330 | 21997 | 62.26% | 60.61% | **20975** | 59.37% | 68.44% |
| Load Balance | 7965 | 5046 | 63.35% | 57.85% | **3656** | 45.90% | 117.9% |
| Geomean | 28082 | 19003 | 67.67% | 47.78% | **16013** | **57.02%** | **75.37%** |

*Table 4.* Solving time of the solver with/without warm start using the predicted initial basis on the unpresolved datasets.

| | default | bipartite | | | tripartiteBMP | | |
|---|---|---|---|---|---|---|---|
| | Time(s) | Time(s) | Ratio | Improv | Time(s) | Ratio | Improv |
| Mirp1 | 9.17 | 5.93 | 64.67% | 54.64% | **4.73** | 51.58% | 93.87% |
| Mirp2 | **21.07** | 24.11 | 114.4% | -12.61% | 22.03 | 104.6% | -4.36% |
| Anonymous | 12.83 | 9.31 | 72.56% | 37.81% | **9.29** | 72.41% | 38.11% |
| Load Balance | 7.21 | 6.10 | 84.60% | 18.20% | **5.14** | 71.29% | 40.27% |
| Geomean | 11.56 | 9.49 | 82.09% | 21.81% | **8.40** | **72.66%** | **37.62%** |

*Table 5.* Number of iterations of the solver with/without warm start using the predicted initial basis on the presolved datasets.

| | default | bipartite | | | tripartiteBMP | | |
|---|---|---|---|---|---|---|---|
| | Iters | Iters | Ratio | Improv | Iters | Ratio | Improv |
| Mirp1 | 22160 | 13715 | 61.89% | 61.57% | **7118** | 32.12% | 211.4% |
| Mirp2 | 40091 | 36528 | 91.11% | 9.75% | **34543** | 86.16% | 16.06% |
| Anonymous | 9452 | 7651 | 80.95% | 23.54% | **7023** | 74.30% | 70.18% |
| Load Balance | 7523 | 4829 | 64.19% | 55.79% | **4681** | 62.57% | 34.59% |
| Geomean | 15854 | 11664 | 73.57% | 35.92% | **9482** | **59.81%** | **67.20%** |

*Table 6.* Solving time of the solver with/without warm start using the predicted initial basis on the presolved datasets.

| | default | bipartite | | | tripartiteBMP | | |
|---|---|---|---|---|---|---|---|
| | Time(s) | Time(s) | Ratio | Improv | Time(s) | Ratio | Improv |
| Mirp1 | 7.12 | 5.22 | 73.31% | 36.40% | **2.99** | 41.99% | 138.1% |
| Mirp2 | 7.91 | 7.54 | 95.32% | 4.91% | **6.56** | 82.93% | 20.57% |
| Anonymous | **0.985** | 1.23 | 124.9% | -19.92% | 1.17 | 118.8% | -15.81% |
| Load Balance | 2.26 | 1.63 | 72.12% | 38.65% | **1.59** | 70.35% | 42.15% |
| Geomean | 3.35 | 2.98 | 88.96% | 12.41% | **2.46** | **73.43%** | **36.18%** |

*Table 7.* Performance on large LP instances.

| | Iters | | Time(s) | |
|---|---|---|---|---|
| | Default | TripartiteBMP | Default | TripartiteBMP |
| LR1_DR04_VC05_V17b_t360 | 561299 | 409428 | 491.48 | 250.70 |
| LR1_DR05_VC05_V25b_t360 | 706451 | 777452 | 474.74 | 513.63 |
| LR1_DR08_VC05_V40b_t180 | 301789 | 309473 | 143.06 | 128.84 |
| LR1_DR08_VC10_V40b_t120 | 267155 | 298875 | 112.76 | 131.60 |
| LR1_DR12_VC10_V70a_t180 | 836561 | 619425 | 751.76 | 606.93 |
| **Geomean** | 484672 | 448942 | 309.27 | 265.74 |
| **Ratio** | 1 | 0.926 | 1 | 0.859 |

presolved datasets respectively, showcasing test precision, recall, accuracy, and basic variable selection accuracy. It is evident that our model holds an advantage in predicting an initial basis that is closer to the optimal solution.

Table 3 and 4 show the impact of the predicted initial basis on accelerating the solver on the unpresolved dataset, and Table 5 and 6 show the one on the presolved dataset. The term "default" refers to the solver's default setting without warm start. "Iters" and "Time" represent the geometric mean of the number of iterations and solving time for each LP instance in the test set. "Ratio" denotes the ratio of the number of iterations using the predicted initial basis to that without warm start. "Improv" indicates the increase in speed compared to the default mode. Our model significantly reduces the number of iterations and solving time, achieving, on average, a twofold speed improvement over SOTA on the unpresolved datasets and a threefold improvement on the presolved datasets. This demonstrates that our model not only provides an initial basis closer to the optimal one but also ensures its high quality for practical use by the solver.

Our approach scales effectively to large LP instances. In the **Mirp2** dataset, some instances take several minutes to several hours to process. As shown in Table 7, we demonstrate the acceleration achieved on these large instances. The computational overhead of our tripartite graph-based model with two bidirectional message-passing steps is comparable to that of the bipartite graph-based model with six message-passing steps, typically taking less than 1 second. As shown in Table 12, increasing the number of message-passing layers in the bipartite graph-based model does not necessarily lead to better prediction quality. Our GNN architecture effectively leverages its complexity, fully utilizing the computational overhead to predict a higher-quality basis. In simpler LP cases, this added overhead may slightly increase the total solving time, including inference time.

However, for more complex or larger problems, the impact of this overhead becomes negligible.

### 4.3. Cross-dataset Evaluation

We use a model trained on one presolved dataset to predict the initial basis for instances from another presolved dataset. The ratio of solver solving time with and without a warm start is shown in Table 8, where each row represents a training set and each column represents a test set. Among the

*Table 8.* The ratio of solver solving time with and without warm start using the predicted initial basis across the presolved datasets.

| | bipartite | | | | tripartiteBMP | | | |
|---|---|---|---|---|---|---|---|---|
| | Mirp1 | Mirp2 | Anonymous | Load Balance | Mirp1 | Mirp2 | Anonymous | Load Balance |
| Mirp1 | - | **0.865** | 1.35 | 1.13 | - | **0.842** | 1.17 | **0.854** |
| Mirp2 | 1.36 | - | 2.43 | 1.52 | 1.04 | - | 1.75 | 1.27 |
| Anonymous | 1.04 | **0.925** | - | 1.01 | **0.813** | **0.850** | - | **0.562** |
| Load Balance | 1.30 | **0.805** | 1.14 | - | 1.24 | **0.856** | 1.76 | - |
| Geomean | 1.22 | **0.864** | 1.55 | 1.20 | 1.02 | **0.849** | 1.53 | **0.847** |

*Table 9.* Solving time of the solver assisted by two GNN models with various training components on the presolved datasets.

| in units of seconds | bipartiteBMP | | | | | tripartiteBMP | | | | |
|---|---|---|---|---|---|---|---|---|---|---|
| | ALL | -B | -M | -P | -BMP | ALL | -B | -M | -P | -BMP |
| Mirp1 | 4.71 | 4.96 | 4.99 | 5.27 | 5.22 | 2.99 | 3.96 | 3.23 | 2.43 | 4.27 |
| Mirp2 | 6.79 | 7.60 | 7.63 | 7.28 | 7.54 | 6.56 | 8.24 | 6.06 | 7.21 | 7.68 |
| Anonymous | 1.13 | 1.18 | 1.53 | 1.17 | 1.23 | 1.17 | 1.17 | 1.06 | 1.19 | 1.36 |
| Load Balance | 1.58 | 1.64 | 1.65 | 1.62 | 1.63 | 1.59 | 1.6 | 1.65 | 1.59 | 1.65 |
| Geomean | **2.75** | 2.92 | 2.92 | 3.13 | 2.98 | 2.46 | 2.80 | 2.42 | **2.40** | 2.93 |

12 cross-dataset evaluations, only 3 cases tested on Mirp2 successfully accelerate the solver using the bipartite graph-based GNN. In contrast, our tripartite graph-based GNN, under comprehensive supervision, accelerates 6 out of the 12 settings. Additionally, our model consistently achieves a better acceleration ratio than the SOTA across all test sets. This demonstrates that our model is more practical for real-world applications, where we need to use a trained model to predict the initial basis for instances from unseen datasets.

### 4.4. Ablation Study

We remove each component from the training process for two GNN models on the presolved datasets to see their role in accelerating solving – see Table 9. We observe that each training component is advantageous in improving performance for the bipartite graph-based GNN. Also, the tripartite GNN exhibits a clear advantage in reducing the iteration count compared to the bipartite GNN with the same training settings. Moreover, the comprehensive supervision can lead to a more significant improvement in the performance of the tripartite graph-based GNN, showcasing its potential.

We observe that the tripartite model without the **P** component outperforms the version with **P**, primarily due to a significant gain on the **Mirp1** dataset. This suggests that while processing the labels to reduce inconsistency helps the GNN learn and achieve better accuracy, it may occasionally result in an initial basis that is closer to optimal but invalid, requiring additional Phase I effort to repair. However, this appears to be an isolated case, as incorporating **P** consistently improves performance on all other datasets.

### 5. Conclusion and Outlook

Based on the analysis of duality in LP, the crucial role of basic variable selection, the quality of the basis in practical tasks, and the inconsistency in raw labels provided by

the solver, we introduce a tripartite graph-based GNN and comprehensive supervision. Significant enhancements have been observed in reducing the number of iterations and solving time of solvers by utilizing the initial basis generated by our model.

As learning for LP is still in its relatively early stage, in the future, we will investigate further factors beyond basis accuracy and infeasibility that influence the solver's actual performance with a warm start. Additionally, we will explore whether the tripartite graph representation remains effective for other LP- and MIP-related tasks.

### Impact Statement

Linear programming has been the central problem in computer science, AI and many other areas for science and engineering. Our technique could potentially accelerate and improve the accuracy of the solvers and thus could have significant impact to the society.

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

# A. Appendix / supplemental material

## A.1. Duality in linear programming

For the general form of a Linear Programming (LP) problem:

$$\min_{x \in \mathbb{R}^n} \quad \boldsymbol{c}^\top \boldsymbol{x} \tag{5}$$

$$s.t. \quad \boldsymbol{l}^s \le \boldsymbol{A}\boldsymbol{x} \le \boldsymbol{u}^s, \tag{6}$$

$$\boldsymbol{l}^x \le \boldsymbol{x} \le \boldsymbol{u}^x, \tag{7}$$

we assign a dual variable $y$ to each constraint (splitting bidirectional constraints into two), as Lagrange multipliers, and formulate the Lagrangian function as:

$$
\begin{aligned}
L(\boldsymbol{x}, \boldsymbol{y}_l^s, \boldsymbol{y}_u^s, \boldsymbol{y}_l^x, \boldsymbol{y}_u^x) =\ & \boldsymbol{c}^\top \boldsymbol{x} - (\boldsymbol{y}_l^s)^\top (\boldsymbol{A}\boldsymbol{x} - \boldsymbol{l}^s) - (\boldsymbol{y}_u^s)^\top (\boldsymbol{u}^s - \boldsymbol{A}\boldsymbol{x}) \\
& - (\boldsymbol{y}_l^x)^\top (\boldsymbol{x} - \boldsymbol{l}^x) - (\boldsymbol{y}_u^x)^\top (\boldsymbol{u}^x - \boldsymbol{x}) \tag{8} \\
=\ & \boldsymbol{x}^\top (\boldsymbol{c} - \boldsymbol{A}^\top \boldsymbol{y}_l^s + \boldsymbol{A}^\top \boldsymbol{y}_u^s - \boldsymbol{y}_l^x + \boldsymbol{y}_u^x) \\
& + (\boldsymbol{l}^s)^\top \boldsymbol{y}_l^s - (\boldsymbol{u}^s)^\top \boldsymbol{y}_u^s + (\boldsymbol{l}^x)^\top \boldsymbol{y}_l^x - (\boldsymbol{u}^x)^\top \boldsymbol{y}_u^x. \tag{9}
\end{aligned}
$$

From 8, in combination with 6 and 7, for any feasible solution $\boldsymbol{x}_{fea}$ and any $\boldsymbol{y}_l^s, \boldsymbol{y}_u^s, \boldsymbol{y}_l^x, \boldsymbol{y}_u^x > 0$, we have:

$$\min_{x \in \mathbb{R}^n} L(\boldsymbol{x}, \boldsymbol{y}_l^s, \boldsymbol{y}_u^s, \boldsymbol{y}_l^x, \boldsymbol{y}_u^x) \le L(\boldsymbol{x}_{fea}, \boldsymbol{y}_l^s, \boldsymbol{y}_u^s, \boldsymbol{y}_l^x, \boldsymbol{y}_u^x) \le \boldsymbol{c}^\top \boldsymbol{x}_{fea}. \tag{10}$$

Let $g(\boldsymbol{y}_l^s, \boldsymbol{y}_u^s, \boldsymbol{y}_l^x, \boldsymbol{y}_u^x) = \min_{x \in \mathbb{R}^n} L(\boldsymbol{x}, \boldsymbol{y}_l^s, \boldsymbol{y}_u^s, \boldsymbol{y}_l^x, \boldsymbol{y}_u^x)$, and taking limits of 10 on the left hand side yields:

$$\max_{\boldsymbol{y}_l^s, \boldsymbol{y}_u^s, \boldsymbol{y}_l^x, \boldsymbol{y}_u^x \ge 0} g(\boldsymbol{y}_l^s, \boldsymbol{y}_u^s, \boldsymbol{y}_l^x, \boldsymbol{y}_u^x) \le \boldsymbol{c}^\top \boldsymbol{x}_{fea}. \tag{11}$$

According to the strong duality theorem, if the original problem has a finite optimal solution $\boldsymbol{x}^*$, then

$$\max_{\boldsymbol{y}_l^s, \boldsymbol{y}_u^s, \boldsymbol{y}_l^x, \boldsymbol{y}_u^x \ge 0} g(\boldsymbol{y}_l^s, \boldsymbol{y}_u^s, \boldsymbol{y}_l^x, \boldsymbol{y}_u^x) = \boldsymbol{c}^\top \boldsymbol{x}^*. \tag{12}$$

Hence, we can transform the original problem into finding the maximum value of the function $g$. From 9, we get:

$$g(\boldsymbol{y}_l^s, \boldsymbol{y}_u^s, \boldsymbol{y}_l^x, \boldsymbol{y}_u^x) = \begin{cases} (\boldsymbol{l}^s)^\top \boldsymbol{y}_l^s - (\boldsymbol{u}^s)^\top \boldsymbol{y}_u^s + (\boldsymbol{l}^x)^\top \boldsymbol{y}_l^x - (\boldsymbol{u}^x)^\top \boldsymbol{y}_u^x, & \boldsymbol{S} = 0, \\ -\infty, & \text{otherwise.} \end{cases} \tag{13}$$

where $\boldsymbol{S} = \boldsymbol{c} - \boldsymbol{A}^\top \boldsymbol{y}_l^s + \boldsymbol{A}^\top \boldsymbol{y}_u^s - \boldsymbol{y}_l^x + \boldsymbol{y}_u^x$. Therefore, the problem of finding the maximum value of the function $g$ can be expressed as:

$$\max_{\boldsymbol{y}_l^s, \boldsymbol{y}_u^s, \boldsymbol{y}_l^x, \boldsymbol{y}_u^x \ge 0} \quad (\boldsymbol{l}^s)^\top \boldsymbol{y}_l^s - (\boldsymbol{u}^s)^\top \boldsymbol{y}_u^s + (\boldsymbol{l}^x)^\top \boldsymbol{y}_l^x - (\boldsymbol{u}^x)^\top \boldsymbol{y}_u^x \tag{14}$$

$$s.t. \quad \boldsymbol{A}^\top (\boldsymbol{y}_l^s - \boldsymbol{y}_u^s) + (\boldsymbol{y}_l^x - \boldsymbol{y}_u^x) = \boldsymbol{c}. \tag{15}$$

## A.2. Message passing on the tripartite graph

Let the initial feature vectors for each node in $V_{primal}$, $V_{dual}$ and $C$ be $\boldsymbol{h}_{x_j^{primal}}^{(0)}$, $\boldsymbol{h}_{x_j^{dual}}^{(0)}$ and $\boldsymbol{h}_{s_i}^{(0)}$, respectively, with the feature vector of the global node $O$ being $\boldsymbol{h}_O^{(0)}$. At the beginning, each initial feature is mapped to a higher dimension

$$
\begin{aligned}
\boldsymbol{h}_O^{(1)} &= \boldsymbol{W}_{01} \boldsymbol{h}_O^{(0)}, \\
\boldsymbol{h}_{x_j^{primal}}^{(1)} &= \boldsymbol{W}_{02} \boldsymbol{h}_{x_j^{primal}}^{(0)}, \\
\boldsymbol{h}_{x_j^{dual}}^{(1)} &= \boldsymbol{W}_{03} \boldsymbol{h}_{x_j^{dual}}^{(0)}, \\
\boldsymbol{h}_{s_i}^{(1)} &= \boldsymbol{W}_{04} \boldsymbol{h}_{s_i}^{(0)}.
\end{aligned}
$$

Then nodes in $V_{primal}$, $V_{dual}$ and $C$ aggregate information to $O$

$$h_O^{(2)} = \sigma((\boldsymbol{W}_{11}\boldsymbol{h}_O^{(1)} + \boldsymbol{W}_{12}\sum_{j\in[n]}[(-c_j)\boldsymbol{h}_{x_j^{primal}}^{(1)l} + c_j\boldsymbol{h}_{x_j^{primal}}^{(1)r}]$$
$$+\boldsymbol{W}_{13}\sum_{j\in[n]}[(-l_j^x)\boldsymbol{h}_{x_j^{dual}}^{(1)l} + c_j^x\boldsymbol{h}_{x_j^{dual}}^{(1)r}] + \boldsymbol{W}_{14}\sum_{i\in[m]}[(-l_i^s)\boldsymbol{h}_{s_i}^{(1)l} + u_i^s\boldsymbol{h}_{s_i}^{(1)r}]),$$

where the feature vectors of each node in $V_{primal}$, $V_{dual}$ and $C$ are split into two halves, with the superscripts $l$ and $r$ indicating the first and second halves of the vectors, respectively. The first half passes information through the lower bounds in the primal or dual problem, while the second half passes through the upper bounds. The activation function $\sigma$, typically a ReLU() function, zeros out negative elements, increasing matrix sparsity. The nodes in those three sets transform and aggregate information using three different learnable matrices $\boldsymbol{W}_{12}$, $\boldsymbol{W}_{13}$ and $\boldsymbol{W}_{14}$ to enhance the global differentiation of information from different components. For each node, the magnitude of information passed to the global node is proportional to its coefficient in the objective function of the primal or dual problem, reflecting the variable's global impact.

Next, information is transmitted from the global node to the nodes in $V_{primal}$, $V_{dual}$ and $C$,

$$h_{x_j^{primal}}^{(2)} = \sigma(\text{CONCAT}(\boldsymbol{W}_{15}\boldsymbol{h}_{x_j^{primal}}^{(1)l} - c_j\boldsymbol{W}_{16}\boldsymbol{h}_O^{(2)}, \boldsymbol{W}_{15}\boldsymbol{h}_{x_j^{primal}}^{(1)r} + c_j\boldsymbol{W}_{16}\boldsymbol{h}_O^{(2)})),$$
$$h_{x_j^{dual}}^{(2)} = \sigma(\text{CONCAT}(\boldsymbol{W}_{17}\boldsymbol{h}_{x_j^{dual}}^{(1)l} - l_j^x\boldsymbol{W}_{18}\boldsymbol{h}_O^{(2)}, \boldsymbol{W}_{17}\boldsymbol{h}_{x_j^{dual}}^{(1)r} + u_j^x\boldsymbol{W}_{18}\boldsymbol{h}_O^{(2)})),$$
$$h_{s_i}^{(2)} = \sigma(\text{CONCAT}(\boldsymbol{W}_{19}\boldsymbol{h}_{s_i}^{(1)l} - l_i^s\boldsymbol{W}_{10}\boldsymbol{h}_O^{(2)}, \boldsymbol{W}_{19}\boldsymbol{h}_{s_i}^{(1)r} + u_i^s\boldsymbol{W}_{10}\boldsymbol{h}_O^{(2)})),$$

where CONCAT represents the vector concatenation function, concatenating feature vectors along the last dimension. Different transformation matrices for the three types of nodes convert the global node's information, ensuring that the information absorbed by each node is proportional to its coefficient in the objective function of the primal or dual problem.

The left half of Fig. 1 is completed at this point. Each node now contains not only its own original information but selectively extracts information from all other nodes in the graph. With more informative node features in hand, bidirectional message passing similar to that in the bipartite graph is carried out.

Before each round of bidirectional message passing, masks are placed on some nodes in $V_{dual}$ and $C$ to remove certain information. For any variable $x_j$, if $l_j^x = -\infty$, then the first half of the feature vector $\boldsymbol{h}_{x_j^{dual}}^{(t-1)l}$ from the previous layer (denoted as the $(t-1)$th layer) of the neural network output is zeroed out, Conversely, if $u_j^x = \infty$, then the second half of the feature vector $\boldsymbol{h}_{x_j^{dual}}^{(t-1)r}$ is set to zero. Similarly, for any constraint variable $s_i$, if $l_i^s = -\infty$, then the first half of the feature vector $\boldsymbol{h}_{s_i}^{(t-1)l}$ from the previous layer of the neural network output is zeroed out, Conversely, if $u_i^s = \infty$, then the second half of the feature vector $\boldsymbol{h}_{s_i}^{(t-1)r}$ is set to zero. Since the infinite constraints in the original problem indicate the absence of these constraints, and these infinite constraints do not actually correspond to variables in dual problems, they need to be masked to ensure they do not participate in any message passing. In the previous global message passing step, these nodes do not have edges connecting them to global nodes and do not participate in message passing. However, in bidirectional message passing, as they are bound together with the other constraints through coefficients in the constraint matrix and connected to other nodes, the masks are used to eliminate their influence on message passing. Denote the feature vectors of the nodes in sets $V_{dual}$ and $C$ in the $t-1$th layer output as $\boldsymbol{h}_{v^{dual}}^{(t-1)'}$ and $\boldsymbol{h}_s^{(t-1)'}$ after applying the masks. The process of bidirectional message passing can be represented as

$$h_{x_j^{primal}}^{(t)} = \sigma(\boldsymbol{W}_{t_1}\boldsymbol{h}_{x_j^{primal}}^{(t-1)} + \boldsymbol{W}_{t_2}\boldsymbol{h}_{x_j^{dual}}^{(t-1)'} + \boldsymbol{W}_{t_3}\sum_{i\in[m]}a_{ij}\boldsymbol{h}_{s_i}^{(t-1)'}),$$
$$h_{x_j^{dual}}^{(t)} = \sigma(\boldsymbol{W}_{t_4}\boldsymbol{h}_{x_j^{dual}}^{(t-1)'} + \boldsymbol{W}_{t_5}\boldsymbol{h}_{x_j^{primal}}^{(t-1)}),$$
$$h_{s_i}^{(t)} = \sigma(\boldsymbol{W}_{t_6}\boldsymbol{h}_{s_i}^{(t-1)'} + \boldsymbol{W}_{t_7}\sum_{j\in[n]}a_{ij}\boldsymbol{h}_{x_j^{primal}}^{(t-1)}),$$

where different learnable transformation matrices $\boldsymbol{W}_{t_2}$ and $\boldsymbol{W}_{t_3}$ are used to aggregate information from $V_{dual}$ and $C$ nodes in the $V_{primal}$ set, respectively, and nodes in $V_{dual}$ and $C$ receive information from $V_{primal}$ through different matrices, $\boldsymbol{W}_{t_5}$ and $\boldsymbol{W}_{t_6}$, respectively.

In practice, it is often preferred to conduct two rounds of bidirectional information exchange after the global information transmission, resulting in $\boldsymbol{h}_{\boldsymbol{v}^{dual}}^{(4)}$, $\boldsymbol{h}_{\boldsymbol{v}^{primal}}^{(4)}$, and $\boldsymbol{h}_{\boldsymbol{s}}^{(4)}$. Among these, the nodes in $V_{dual}$ do not contribute to the final output. The features of all nodes in $V_{primal}$, denoted by $\boldsymbol{h}_{\boldsymbol{v}^{primal}}^{(4)}$, are processed through a linear layer to map them to a three-dimensional space, providing the logits for each variable. Similarly, the features obtained from $C$ nodes, after masking, are passed through another linear layer to transform them into a three-dimensional space, yielding the logits for each constraint.

The rationale behind opting for two rounds of bidirectional information exchange is that this allows for the aggregation of a significant portion of the information implied by the constraint matrix. The masking and activation functions essentially sparsify the information within the matrix, and we disregard the sparse effects induced by the masking and activation functions in our analysis. Taking $\boldsymbol{h}_{\boldsymbol{v}^{primal}}^{(4)}$ as an example, it can be represented as a function of $\boldsymbol{h}_{\boldsymbol{v}^{primal}}^{(2)}$, $\boldsymbol{h}_{\boldsymbol{v}^{dual}}^{(2)}$, and $\boldsymbol{h}_{\boldsymbol{s}}^{(2)}$ as

$$
\begin{aligned}
\boldsymbol{h}_{x_j^{primal}}^{(4)} =\ & \boldsymbol{W}_{4_1}\boldsymbol{h}_{x_j^{primal}}^{(3)} + \boldsymbol{W}_{4_2}\boldsymbol{h}_{x_j^{dual}}^{(3)} + \boldsymbol{W}_{4_3}\sum_{i\in[m]}a_{ij}\boldsymbol{h}_{s_i}^{(3)} \\
=\ & \boldsymbol{W}_{4_1}(\boldsymbol{W}_{3_1}\boldsymbol{h}_{x_j^{primal}}^{(2)} + \boldsymbol{W}_{3_2}\boldsymbol{h}_{x_j^{dual}}^{(2)} + \boldsymbol{W}_{3_3}\sum_{i\in[m]}a_{ij}\boldsymbol{h}_{s_i}^{(2)}) \\
& + \boldsymbol{W}_{4_2}(\boldsymbol{W}_{3_4}\boldsymbol{h}_{x_j^{dual}}^{(2)} + \boldsymbol{W}_{3_5}\boldsymbol{h}_{x_j^{primal}}^{(2)}) \\
& + \boldsymbol{W}_{4_3}\sum_{i\in[m]}a_{ij}(\boldsymbol{W}_{3_6}\boldsymbol{h}_{s_i}^{(2)} + \boldsymbol{W}_{3_7}\sum_{k\in[n]}a_{ik}\boldsymbol{h}_{v_k^{primal}}^{(2)}) \\
=\ & (\boldsymbol{W}_{4_1}\boldsymbol{W}_{3_1} + \boldsymbol{W}_{4_2}\boldsymbol{W}_{3_5})\boldsymbol{h}_{x_j^{primal}}^{(1)} + (\boldsymbol{W}_{4_1}\boldsymbol{W}_{3_2} + \boldsymbol{W}_{4_2}\boldsymbol{W}_{3_4})\boldsymbol{h}_{x_j^{dual}}^{(1)} \\
& + (\boldsymbol{W}_{4_1}\boldsymbol{W}_{3_3} + \boldsymbol{W}_{4_3}\boldsymbol{W}_{3_6})\sum_{i\in[m]}a_{ij}\boldsymbol{h}_{s_i}^{(2)} \\
& + \boldsymbol{W}_{4_3}\boldsymbol{W}_{3_7}\sum_{i\in[m]}\sum_{k\in[n]}a_{ij}a_{ik}\boldsymbol{h}_{v_k^{primal}}^{(2)}\quad,
\end{aligned}
$$

Following this analysis, we observe that the feature of a node $x_j$ in $\boldsymbol{V}_{primal}$ consists of message from the node in $\boldsymbol{V}_{dual}$ corresponding to its upper and lower bounds, the nodes in $C$ corresponding to constraints it lies in, and some other variable nodes in $\boldsymbol{V}_{primal}$. The message passed from a constraint node $s_i$ in $C$ is proportional to the coefficients $a_{ij}$ corresponding to the variable in the respective constraint, and is positively correlated to any change $\delta$ in the constraint that influences the total value of the variable, $a_{ij}\delta$, in the dual problem. Moreover, the message passed from a variable node $v_k$ in $\boldsymbol{V}_{primal}$ is associated with $\sum_{i\in[m]}\sum_{k\in[n]}a_{ij}a_{ik}$, reflecting the inner product between the column vectors corresponding to variables $x_j$ and $v_k$ in the constraint matrix, which is proportional to the fluctuation in the projection of the change $\delta$ in the variable $v_k$ on $A_{:,j}$, which is $\delta\langle A_{:,j}, A_{:,k}\rangle/|A_{:,j}|^2$. Analogously, similar effects of information transfer can be identified for $\boldsymbol{h}_{\boldsymbol{s}}^{(4)}$ from a dual perspective. Consequently, this dual bidirectional information exchange on the bipartite graph effectively consolidates the interactive information between constraints and variables inherent in the LP problem, exhibiting interpretability to a certain extent. Furthermore, some high-order interaction information is captured through the initial global message passing.

### A.3. Analysis on the labels given by LP solvers

For example, consider a simple LP problem as follows:

$$
\begin{aligned}
\min\quad & -x_1 - x_2 \\
s.t.\quad & 0 \leq x_1 + x_2 \leq 1 \\
& 0 \leq x_1, x_2 \leq 2,
\end{aligned}
$$

This problem involves two variables $x_1$ and $x_2$ and one constraint $s_1$. The optimal solution with the 2-norm minimum is $x_1 = 0.5, x_2 = 0.5$ with $s_1 = 1$. The optimal solution obtained through the simplex method could be $x_1 = 1, x_2 = 0, s_1 = 1$ or $x_1 = 0, x_2 = 1, s_1 = 1$. It can be observed that the values of $x_1$ and $x_2$ are always unequal. Due to the equivalence of $x_1$ and $x_2$ in the original problem, no matter how the neural network is constructed, the output for nodes corresponding to $x_1$ and $x_2$ will be the same. Therefore, even for such a simple LP problem, the regression problem of predicting the optimal solution with the simplex method cannot achieve close to 100% accuracy. Additionally, in any case, $s_1$ serves as a non-basic

variable at the upper bound, while either $x_1$ or $x_2$ serves as a basic variable and the other as a non-basis variable at the lower bound. Hence, the labels for $x_1$ and $x_2$ in the basis prediction classification problem will never be the same, leading to at least one variable's prediction being incorrect, resulting in a maximum prediction accuracy of 50% for the basic variables.

This inconsistency in labels arising from the structure of the problem is commonly found in practical LP problems. It stems from the fact that the representation of LP problems is not compact enough and contains redundant information. For instance, $x_1$ and $x_2$ are completely equivalent in the example above and could be combined into a single variable to eliminate the label inconsistency mentioned before. It is noted that advanced solvers often preprocess LP problems by removing redundant information to obtain a more concise problem representation. The runtime for LP problem presolving is usually very short and negligible compared to actual solving time. By inputting the equivalent problem obtained after presolving into the solver, significant computational cost savings can be achieved compared to solving the original problem directly. Consequently, there is limited practical significance in directly using the original problem for basis prediction, as it does not take into account the presolving process. Moreover, as shown in the analysis above, presolving can aid in basis prediction by reducing label inconsistency, improving model fitting, and enhancing basis prediction accuracy. Therefore, utilizing the LP problem obtained after presolving as the dataset not only enhances the learning efficacy of neural networks but also aligns with the standard solving process in practical applications.

Apart from the label inconsistency stemming from the problem structure itself, there can also be label inconsistencies in the optimal basis labels output by the solver. Adding a constraint and a variable to the previous LP problem yields another LP problem without redundant information as follows:

$$
\begin{aligned}
\min \quad & -x_1 - x_2 - x_3 \\
s.t. \quad & x_1 + x_2 = 1 \\
& x_2 + x_3 = 1 \\
& 0 \le x_1, x_2, x_3 \le 2,
\end{aligned}
$$

This problem involves three variables $x_1, x_2, x_3$ and two constraints $s_1, s_2$. The optimal solution with the 2-norm minimum is $x_1 = 1, x_2 = 0, x_3 = 1$ with $s_1 = 1, s_2 = 1$, which is also the optimal solution obtained by the simplex method. The prediction values of neural networks can approximate the solution obtained by the simplex method with nearly 100% accuracy. However, this optimal solution corresponds to multiple optimal bases. One possible labeling scheme includes $\text{Label}_{x_1} = 1, \text{Label}_{x_2} = 0, \text{Label}_{x_3} = 1, \text{Label}_{s_1} = 0, \text{Label}_{s_2} = 2$, indicating that $x_1$ and $x_3$ are basic variables, $x_2$ is a non-basic variable set to the lower bound 0, and $s_1$ and $s_2$ correspond to the lower bound 1 in one constraint and upper bound 1 in the other. In this labeling, the labels of the two constraints $s_1$ and $s_2$ are different, yet in the LP problem, $x_1$ and $x_3$ are completely symmetrical and so are the two constraints $s_1$ and $s_2$, implying that the nodes in the graph representation are equivalent, leading to the neural network outputting the same values for them, at least one of which is different from their labels. Therefore, in this scenario, the prediction accuracy of the constraint basis variables can reach a maximum of only 50%, even though the prediction accuracy for the optimal solution can be 100%.

The mentioned label inconsistency primarily arises when the upper and lower bounds of variables or constraints are equal. When the upper and lower bounds of variables are equal, indicating a fixed value for the variable, it can be eliminated through presolving. However, when the upper and lower bounds of constraints are equal, representing an equality constraint, these constraints cannot be removed during presolving. Therefore, additional processing is required to address the label inconsistency corresponding to the constraints.

### A.4. Datasets

Table 10 and 11 present statistical information for four datasets before and after presolve. 'nMIP' denotes the number of original MIP instances, while 'nLP' represents the number of remaining presolved LP instances, which constitute our final datasets. 'nTrain', and 'nTest' denote the number of LP instances in the training set and in the test set, respectively. 'dens', 'nvar', and 'ncons' refer to the density of constraint matrices, the number of variables, and the number of constraints. They are presented in the 'mean $\pm$ standard deviation' format.

### A.5. Model Setup

We employ the Adam optimizer with a weight decay of $1 \times 10^{-4}$. The number of training epochs is 800. The initial learning rate is configured to 0.001, and the learning rate is reduced by a factor of $\gamma = 0.1$ every 200 epochs. Random seed is set to 0.

*Table 10.* Statistics of the unpresolved datasets.

|  | nMIP | nLP | nTrain | nTest | dens | nvar | ncons |
|---|---|---|---|---|---|---|---|
| Mirp1 | 28 | 28 | 19 | 9 | $2.0\text{e-}4_{\pm 1.0\text{e-}4}$ | $28738_{\pm 24508}$ | $28214_{\pm 24740}$ |
| Mirp2 | 68 | 68 | 47 | 21 | $5.6\text{e-}4_{\pm 4.4\text{e-}4}$ | $33461_{\pm 39425}$ | $10589_{\pm 11365}$ |
| Anonymous | 118 | 76 | 53 | 23 | $4.3\text{e-}4_{\pm 4.9\text{e-}4}$ | $50891_{\pm 39009}$ | $64396_{\pm 58839}$ |
| Load Balancing | 100 | 100 | 70 | 30 | $9.2\text{e-}5_{\pm 1.4\text{e-}6}$ | $61000_{\pm 0}$ | $64305_{\pm 54}$ |

*Table 11.* Statistics of the presolved datasets.

|  | nMIP | nLP | nTrain | nTest | dens | nvar | ncons |
|---|---|---|---|---|---|---|---|
| Mirp1 | 28 | 28 | 19 | 9 | $2.6\text{e-}4_{\pm 1.4\text{e-}4}$ | $26043_{\pm 22772}$ | $24028_{\pm 21581}$ |
| Mirp2 | 68 | 62 | 43 | 19 | $6.9\text{e-}4_{\pm 5.1\text{e-}4}$ | $20351_{\pm 17293}$ | $7126_{\pm 5262}$ |
| Anonymous | 118 | 76 | 53 | 23 | $1.5\text{e-}3_{\pm 6.7\text{e-}4}$ | $7720_{\pm 4406}$ | $5670_{\pm 2437}$ |
| Load Balancing | 100 | 100 | 70 | 30 | $6.7\text{e-}3_{\pm 3.4\text{e-}5}$ | $4014_{\pm 60}$ | $7136_{\pm 119}$ |

The initial features of variable and constraint nodes in the graph are identical to those in the study (Fan et al., 2023), represented as 8-dimensional vectors. The additional global node features in the tripartite graph are composed of the mean and variance of the coefficients of normalized LP problem's objective function, variable bounds, and constraint bounds, forming a 6-dimensional vector. The feature vectors of all nodes in the hidden layers have a dimensionality of 256.

Our tripartite-based GNN model involves one linear layer that projects the initial features into 128-dimensional embeddings, one global message passing iteration between the global node and other nodes, two bidirectional message passing iterations among the remaining nodes, and a final linear layer. In particular, when applying label preprocessing, we choose $\theta = 0.9$ and $N = 10$ to collect the feasible bases in the solving path, obtaining multi-level labels. The weight $\mu_k$ for the $k$-th level label is set as

$$\mu_k = \begin{cases} 0.5^k, & 0 \le k \le 2 \\ 0.25 \times 0.75^{k-2}, & 3 \le k \le 9. \end{cases} \tag{16}$$

On the other hand, the bipartite-based GNN model (Fan et al., 2023) comprises $N$ bidirectional message passing iterations between variable nodes and constraint nodes and a final linear layer. The default setting for $N$ is 2, and we also experiment with models using $N = 4$, $N = 6$, and $N = 8$. As shown in Table 12, for both models utilizing the training process from (Fan et al., 2023), labeled as 'bipartite', and our proposed training process, labeled as 'bipartiteBMP', adding more layers results in little overall improvement. Therefore, using the tripartite graph-based representation, increasing the number of layers does not enhance the learning ability. For this reason, we choose $N = 2$ as the setting for the comparison in the experiments section.

We also conduct experiments studying the influence of the number of bidirectional layers on the final solver speedup on three of the unpresolved datasets. To better study the graph representation itself, we isolate the influence of training settings by using the base training without a loss function for basic variable selection, the loss function for feasibility, and label preprocessing. As shown in Table 13, bidirectional message passing is crucial, as it effectively utilizes the coefficients in the constraint matrix. Additionally, adding more bidirectional message passing layers results in a slight improvement.

We compare bipartite GNN and tripartite GNN using TransformerConv architectures, as well as tripartite GNN using GraphConv, on three of the unpresolved datasets. As shown in Table 14, our tripartite representation demonstrates a modest advantage over the bipartite baseline. Additionally, we observe that GraphConv generally outperforms TransformerConv, further supporting our claim that the GraphConv architecture aligns well with the structure of LP problems and is sufficiently expressive to capture the necessary properties.

*Table 12.* Iteration counts across bipartite GNNs with different number of bidirectional message passing layers

|  | bipartite | | | | bipartiteBMP | | | |
|---|---|---|---|---|---|---|---|---|
|  | N=2 | N=4 | N=6 | N=8 | N=2 | N=4 | N=6 | N=8 |
| Mirp1 | 13715 | 12085 | 13272 | 15817 | 10259 | 12167 | 9949 | 10618 |
| Mirp2 | 36528 | 36449 | 38757 | 39676 | 32302 | 33979 | 35803 | 35400 |
| Anonymous | 7651 | 9381 | 7717 | 8226 | 6891 | 7990 | 7728 | 7131 |
| Load Balancing | 4829 | 4624 | 4645 | 3956 | 4754 | 4602 | 4457 | 3737 |
| Geomean | 11664 | 11757 | 11653 | 11954 | 10207 | 11103 | 10525 | 10004 |

*Table 13.* Iteration counts and solving time across tripartite GNNs with different number of bidirectional message passing layers

|  | Iters | | | | Time(s) | | | |
|---|---|---|---|---|---|---|---|---|
|  | N=0 | N=2 | N=4 | N=6 | N=0 | N=2 | N=4 | N=6 |
| Mirp | 18798 | 14017 | 12109 | 10813 | 8.29 | 7.43 | 7.07 | 6.39 |
| Anonymous | 60146 | 23583 | 23166 | 21342 | 7.19 | 7.12 | 6.75 | 6.87 |
| Load_balancing | 4070 | 4357 | 4291 | 4512 | 16.71 | 10.49 | 10.98 | 8.85 |
| **Geomean** | 16633 | 11293 | 10637 | 10136 | 9.99 | 8.22 | 8.06 | 7.30 |

*Table 14.* Comparison of iteration counts across different GNN architectures

|  | Bipartite-TransformerConv | | Tripartite-TransformerConv | | Tripartite-GraphConv | |
|---|---|---|---|---|---|---|
|  | Iters | Time(s) | Iters | Time(s) | Iters | Time(s) |
| Mirp | 16022 | 7.24 | 16146 | 6.96 | 14017 | 6.04 |
| Anonymous | 4561 | 9.87 | 4804 | 9.75 | 4357 | 5.74 |
| Load_balancing | 23946 | 6.37 | 21124 | 6.44 | 23583 | 10.61 |
| **Geomean** | 12051 | 7.69 | 11790 | 7.59 | 11294 | 7.17 |

