# OpenReview forum: "Learning Initial Basis Selection for Linear Programming via Duality-Inspired Tripartite Graph Representation and Comprehensive Supervision"
_ICML.cc/2025/Conference — ICML 2025 poster_

### Official Review · Reviewer_SeCg · 2025-03-09

**Overall Recommendation:** 4

**Summary:**

Linear Programming (LP) is fundamental to numerous real-world applications, driving significant investment and research into improving the Simplex method, a widely used algorithm for solving LPs. Over decades, various heuristics have been developed to enhance solver efficiency, one of which is choosing an optimal initial basis, a critical factor influencing solver performance.

In this paper, the authors propose a novel framework that leverages a new way of representing LPs and a specialized loss function to predict a superior starting point for the Simplex algorithm. Their approach is validated through extensive experiments, demonstrating state-of-the-art (SOTA) performance improvements over existing heuristics, thereby showcasing its potential to accelerate LP solvers effectively.

**Claims And Evidence:**

Yes. The evidence supports the claims. I have some clarifying questions. Please see the sections below.

**Essential References Not Discussed:**

It looks good.

**Experimental Designs Or Analyses:**

Experimental design looks good. All ablations are clearly showing the efficacy of their proposal.

I have a few proposals for an additional ablation for better understanding

 -  $L_{multi}$ - How are $\theta$ and $\mu_k$ selected? Did the authors do some hyperparameter tuning on these parameters? Perhaps, a simple ablation would be to do random sampling and compare that with the chosen one.

 - Tripartite representation has also been well explained. However, just like how the effect of the number of message passing round is understood in Table 11 (in Appendix), it will be useful to have such a table for tripartite representation. I am particularly interested in what power does N=0 (i.e., only the messages passed back and forth from the global node).

There is a knawing worry that the improvements might just be because of the increase in the number of parameters. It will be useful to have details around

- The number of learnable parameters in bipartite vs tripartite representation
- For all the datasets, geometric mean of the number of nodes and edges in bipartite vs tripartite representation
- Comparison of runtime of a single pass between bipartite vs tripartite representation

**Methods And Evaluation Criteria:**

Overall methods are clear and well explained. I have a few questions.
- Fan et al. uses LIBSVM and STOCH datasets. Can the authors display their results on these two datasets as well?
- Fan et al. establish that their method is better than CPLEX. Therefore, it doesn’t seem necessary to compare the proposal with CPLEX. -However, since Gurobi is also a widely accepted solver for LPs, is it possible to have Gurobi as one of the baselines (time or iterations)? This will be useful in completing the picture of Initial basis selection problem and standards.

**Other Comments Or Suggestions:**

- The notation in Basis section of page 2 is wrong. $B_x$ should be in the subscript. In addition, it should be told explicitly that [ ] means concatenation.
- $x_{N_x}$ is wrongly placed inside the second bracket

**Other Strengths And Weaknesses:**

Authors proposal is quite novel and the results show substantial improvement over the benchmarks. I have suggestions for adding a baseline and showcasing the performance on more datasets as used by their predecessor work. Additional suggestion on ablation will also make for a better work.

**Questions For Authors:**

- What is the form of $l$ in Equation 4?
- It will be useful to understand when tripartite and bipartite representation might look similar. Can the authors take a few examples such as  $l_x=0$, $u_x=\inf$, $l_s=-\inf$, $u_s=b$? Some examples that can help understand how representations might differ wildly under two cases.


In general, I am optimistic about the proposal. I am inclined to increase my rating once I hear the authors' rebuttal.

**Relation To Broader Scientific Literature:**

The authors’ proposal for tripartite representation of LPs for GNN is novel. This could actually be used across several other deep learning-based heuristic design for solvers as well as other deep learning-aided combinatorial optimization problems.

Their work falls under the broader theme of making LP solvers faster using deep learning methods.

Their work has also identified core issues in dealing with predictions in LP such as label inconsistencies, which haven’t been addressed in the literature.

Corresponding to the specific problem of Initial Basis Selection, the authors improve upon their predecessor and report substantial improvements.

**Theoretical Claims:**

N/A

---

> ### Author Rebuttal · Authors · 2025-04-01
>
> ### Fan et al. uses LIBSVM and STOCH datasets. Can the authors display their results on these two datasets as well?
> > It would certainly be helpful to compare results across all datasets from the original work. However, since these two datasets were generated by the authors and are not publicly available, we are unable to reproduce them. Instead, we incorporate other publicly available datasets to expand our evaluation.
>
> ### It will be useful to have such a table (Table 11 in Appendix)) for tripartite representation. I am particularly interested in what power does N=0 (i.e., only the messages passed back and forth from the global node).
>
> > We add experiments on using tripartite graph-based GNN with difference bidirectional message passing layers (0, 2, 4 and 6 respectively) compared to the default solving mode without warm start. We use the base training without loss function for basic variable selection, the loss function for feasibility, and label preprocessing. Here are the number of iteration and solving time on the unpresolved datasets:
> > Iteration number:
> >
> | datasets       | n=0   | n=2   | n=4   | n=6   | default |
> | -------------- | ----- | ----- | ----- | ----- | ---- |
> | Mirp           | 18798 | 14017 | 12109 | 10813 | 25432 |
> | Anonymous      | 60146 | 23583 | 23166 | 21342 | 35330 |
> | Load_balancing | 4070  | 4357  | 4291  | 4512  | 7965 |
> | geomean        | 16633 | 11293 | 10637 | 10136 | 19271 |
>
> > Solving time in seconds:
> >
> | datasets       | n=0   | n=2   | n=4   | n=6   | default |
> | -------------- | ----- | ----- | ----- | ----- | ---- |
> | Mirp           | 8.29 | 7.43 | 7.07 | 6.39 | 11.17 |
> | Anonymous      | 7.19 | 7.12 | 6.75 | 6.87 | 9.08 |
> | Load_balancing | 16.71  | 10.49  | 10.98  | 8.85  | 7965 |
> | geomean        | 9.99 | 8.22 | 8.06 | 7.30 | 11.39 |
>
> > Due to time constraints, we have not yet completed testing on the Mirp2 dataset, but we will provide the results later. It is evident that bidirectional message passing is crucial as it effectively utilizes the coefficients in the constraint matrix. Additionally, adding more bidirectional message passing layers results in a slight improvement.
>
> ### There is a knawing worry that the improvements might just be because of the increase in the number of parameters.
> > The computational overhead of our tripartite graph-based model with two bidirectional message-passing steps is comparable to that of the bipartite graph-based model with six message-passing steps. As shown in Table 11 of the appendix, increasing the number of message-passing layers in the bipartite graph-based model does not necessarily improve prediction quality. Our GNN architecture effectively leverages its complexity, making full use of the computational overhead to predict a higher-quality basis.
>
> ### What is the form of $L$ in Equation 4?
> ### How are  $\theta$ and  $\mu_k$ selected? Did the authors do some hyperparameter tuning on these parameters?
>
> > $\mu_k$ is the weight of the loss calculated from the $k^{th}$ label using Equation 2, which defines the weighted cross-entropy. The value of $\mu_k$ is given in Equation 16 and is determined based on experience rather than precise fine-tuning. It is worth investigating the hyperparameter settings for $\theta$, $k$, and $\mu_k$.
>
> ### It will be useful to understand when tripartite and bipartite representation might look similar. Can the authors take a few examples such as $l_x=0$, $u_x=inf$, $l_s=-inf$, $u_s=b$? Some examples that can help understand how representations might differ wildly under two cases.
> > For LP problems in standard form:
> \begin{equation}
> \begin{aligned}
>     \min_{\boldsymbol{x} \in \mathbb{R}^n} \quad & \boldsymbol{c}^\top \boldsymbol{x} \\\\
>     \text{s.t.} \quad \boldsymbol{A} \boldsymbol{x} &\leq \boldsymbol{b}, \\\\
>     \boldsymbol{x} &\geq 0,
> \end{aligned}
> \end{equation}
>
> > we have $l_x = 0$, $u_x = \infty$, $l_s = -\infty$, $u_s = b$. In the tripartite GNN, the initial message passing occurs only between the global node and nodes in $V_{primal}$ (see Figure 1(a)), which differs from the bipartite GNN. The other difference is that the tripartite GNN introduces additional message passing between $V_{dual}$ and $V_{primal}$ via indentity edges—an interaction that may be redundant in this case. As a result, the bipartite GNN could be preferable since it maintains a similar structure to the tripartite GNN while incurring lower computational overhead.
>
> > However, for LP problems in a more general form, as shown in Equation (1), where $l_x$, $u_x$, $l_s$, and $u_s$ are all finite numbers, the initial message passing between the global node and other nodes introduces richer information into $V_{dual}$, $V_{primal}$ and $C$. Subsequent message passing between $V_{dual}$, $V_{primal}$ and $C$ can then fully exploit these enriched node embeddings. In this scenario, the tripartite representation differs significantly from the bipartite one, allowing the tripartite GNN to exhibit greater expressiveness.

---

### Official Review · Reviewer_RwiH · 2025-03-13

**Overall Recommendation:** 4

**Summary:**

The paper proposes a novel approach for selecting an initial basis for linear programming (LP) solvers using a duality-inspired tripartite graph neural network (GNN).
The following are the three main contributions:
- A tripartite graph representation for LP problems inspired by duality theory, which enhances feature extraction and GNN expressiveness
- Novel loss functions targeting basic variable selection and basis feasibility, along with multi-level labels from the solving path
- Data preprocessing schemes to address label inconsistencies in solver-derived data

The approach significantly outperforms state-of-the-art methods in predicting initial basis with higher accuracy, reducing the number of iterations and solving time in LP solvers.

## update after rebuttal
I thank the authors for the detailed rebuttal, their response sufficiently addresses my concerns, hence I am updating my scores.

**Claims And Evidence:**

The claims in the paper are supported by convincing evidence. The authors demonstrate that the proposed approach
- outperforms the state-of-the-art bipartite model across datasets.
- reduces iterations for the previous state-of-the-art when used for a warm start in the HiGHS LP solver.
- they also include ablation studies to demonstrate the value of each component (basic variable selection, feasibility loss, and preprocessing)

**Essential References Not Discussed:**

While the paper covers most relevant literature, they don't discuss neural network based approaches for predicting optimal solutions to optimization problems beyond just basis selection.

**Experimental Designs Or Analyses:**

Refer to Methods and Evaluation Criteria

**Methods And Evaluation Criteria:**

The authors evaluate their approach on four well-known datasets (Mirp1, Mirp2, Anonymous, and Load Balance) from MIP problems relaxed to LP and compare it with the state-of-the-art bipartite GNN model (Fan et al., 2023). In addition, they perform a detailed ablation study to demonstrate the utility of each of the proposed approach's core components.

The experiments are well-designed to test the paper's key claims, with appropriate metrics to evaluate prediction quality and performance (iterations).

**Other Comments Or Suggestions:**

Refer to other sections

**Other Strengths And Weaknesses:**

Weaknesses
- limited discussion of computational overhead introduced by the more complex GNN architecture
- it is unclear how the proposed approach will scale to very large LP instances beyond those in the test datasets

**Questions For Authors:**

- How does the computational cost of the tripartite GNN compare to the bipartite model, and how does this balance with the solver time savings? It would be valuable to include runtime comparison of the GNN inference time versus the time saved in the solving process
- What are the limitations of your approach for very large-scale LP problems?

**Relation To Broader Scientific Literature:**

The authors acknowledge prior work in these areas and clearly articulate how their approach extends beyond previous methods. The contribution of the paper addresses the specific gaps in the literature in the intersection of deep learning and mathematical optimization.
- the paper extends the bipartite graph representation of optimization problems
-  builds upon on work for tripartite graph representations and extends it to handle general LP form
- addresses a fundamental issue in learning-based optimization

**Theoretical Claims:**

The claims are generally well-supported with mathematical formulations and examples, particularly regarding duality theory and the message-passing mechanism in the tripartite graph.

---

> ### Author Rebuttal · Authors · 2025-04-01
>
> ### limited discussion of computational overhead introduced by the more complex GNN architecture
> ### How does the computational cost of the tripartite GNN compare to the bipartite model, and how does this balance with the solver time savings? It would be valuable to include runtime comparison of the GNN inference time versus the time saved in the solving process
>
> > The computational overhead of our tripartite graph-based model with two bidirectional message-passing steps is comparable to that of the bipartite graph-based model with six message-passing steps, typically taking less than 1 second. As shown in Table 11 of the appendix, increasing the number of message-passing layers in the bipartite graph-based model does not necessarily lead to better prediction quality. Our GNN architecture effectively leverages its complexity, fully utilizing the computational overhead to predict a higher-quality basis. In simpler LP cases, this added overhead may slightly increase the total solving time, including inference time. However, for more complex or larger problems, the impact of this overhead becomes negligible.
>
> ### unclear how the proposed approach will scale to very large LP instances beyond those in the test datasets
> ### What are the limitations of your approach for very large-scale LP problems?
>
> > Our approach scales effectively to large LP instances. In the Mirp2 dataset, some instances are so large that they require several minutes to several hours for the solver to process. We demonstrate the acceleration achieved on these large instances as follows:
> >
> | instances               | iter_default | iter_tripartiteBMP | time_default | time_tripartiteBMP |
> | ----------------------- | ------------ | ------------------ | ------------ | ------------------ |
> | LR1_DR04_VC05_V17b_t360 | 561299       | 409428             | 491.48       | 250.7              |
> | LR1_DR05_VC05_V25b_t360 | 706451       | 777452             | 474.74       | 513.63              |
> | LR1_DR08_VC05_V40b_t180 | 301789       | 309473             | 143.06       | 128.84              |
> | LR1_DR08_VC10_V40b_t120 | 267155       | 298875             | 112.76       | 131.6              |
> | LR1_DR12_VC10_V70a_t180 | 836561       | 619425             | 751.76       | 606.93              |
> | geomean | 484672       | 448942             | 309.27       | 265.74              |
> > The iter_* columns represent the number of iterations, while the time_* columns indicate the solving time in seconds. In practice, the GNN model is particularly useful for large LP problems, as its computational overhead is negligible compared to the overall solving time. However, for extremely large LPs—which translate to massive graphs—memory constraints may require the use of graph sampling techniques like GraphSAGE during both training and inference.

---

### Official Review · Reviewer_BL5r · 2025-03-14

**Overall Recommendation:** 3

**Summary:**

This paper proposes a GNN-based approach for learning initial basis selection in LP, aiming to accelerate the simplex method. Inspired by LP duality, the authors introduce a tripartite graph representation to better capture problem structure. Additionally, they design new loss functions to improve basic variable selection and basis feasibility, along with data preprocessing schemes to reduce label inconsistencies.

## Update after rebuttal

I thank the authors for their thoughtful and detailed rebuttal. The clarifications regarding the distinction between basis accuracy and solver acceleration were particularly helpful and addressed my earlier confusion about the paper’s core claims.

I also appreciate the additional experiments and analysis related to the tripartite graph representation and the choice of GNN architecture. The authors not only provided ablation studies with varying message passing depths but also included comparisons with Graph Transformer architectures. Their justification for using the GraphConv-based model — highlighting interpretability, computational efficiency, and consistency with related work — is convincing and well-supported by the new results.

Given these clarifications and improvements, I believe the authors have sufficiently addressed my concerns. I have therefore increased my score to a positive recommendation.

**Claims And Evidence:**

While the paper presents an interesting method, the writing is sometimes contradictory, leading to confusion about the claims:

- The abstract states, "a closer initial basis does not always result in greater acceleration," but later mentions "achieving high prediction accuracy." It is unclear what accuracy refers to in this context—closeness to the optimal basis or actual solver acceleration.
- Similarly, the introduction states, "A better starting point may be closer to the potential optimal solution in terms of logical pivot distance, often resulting in fewer simplex iterations," which contradicts the earlier claim that closeness does not always lead to acceleration.
- Clarifying these statements would improve the paper’s logical consistency and ensure a clearer understanding of the contributions.

**Essential References Not Discussed:**

No major missing references, but the paper could compare its approach to other LP warm-starting methods that do not use GNNs.

**Experimental Designs Or Analyses:**

The experiments primarily compare the tripartite GNN to a single bipartite GNN model. To strengthen the evaluation:

- More GNN architectures should be tested to confirm the robustness of the tripartite representation.
- The impact of each proposed modification (graph representation, loss functions, preprocessing) should be evaluated separately to better understand their individual contributions.

**Methods And Evaluation Criteria:**

The paper proposes a tripartite graph representation to replace the standard bipartite graph representation used in LP-based GNN models. This is a reasonable design choice, but the experimental validation is limited:
- The tripartite graph should be tested against multiple GNN architectures (not just one) to demonstrate its effectiveness across different models.
- The impact of the tripartite representation should be better isolated from other factors (e.g., loss functions, preprocessing).

**Other Comments Or Suggestions:**

None

**Other Strengths And Weaknesses:**

Strengths
- The tripartite graph representation is a novel adaptation based on LP duality.
- The method shows strong experimental improvements over the SOTA bipartite GNN model.
- The data preprocessing steps help address label inconsistencies in LP solvers.

Weaknesses
- Potential desk rejection risk: The paper omits required formatting elements (e.g., line numbers, "Anonymous Authors"), which could lead to desk rejection.
- Logical inconsistencies: The discussion on closeness of the initial basis vs. solver acceleration is contradictory.
- Limited experimental diversity: The tripartite graph representation should be tested on multiple GNN architectures to confirm its general effectiveness.

**Questions For Authors:**

See Weaknesses. I find the proposed method interesting and promising, but the writing inconsistencies, formatting errors, and limited experimental validation raise concerns. I am assigning a borderline score for now but am open to raising my score if the authors provide strong clarifications and additional experiments in their response.

**Relation To Broader Scientific Literature:**

The paper correctly cites LP and MIP learning-based methods, but it would benefit from a broader discussion on alternative warm-starting techniques beyond GNNs.

**Theoretical Claims:**

The theoretical motivation for the tripartite representation is reasonable, as it leverages LP duality.

---

> ### Author Rebuttal · Authors · 2025-04-01
>
> ### The abstract states, "a closer initial basis does not always result in greater acceleration," but later mentions "achieving high prediction accuracy." It is unclear what accuracy refers to in this context—closeness to the optimal basis or actual solver acceleration.
> ### Similarly, the introduction states, "A better starting point may be closer to the potential optimal solution in terms of logical pivot distance, often resulting in fewer simplex iterations," which contradicts the earlier claim that closeness does not always lead to acceleration.
> ### Clarifying these statements would improve the paper’s logical consistency and ensure a clearer understanding of the contributions.
>
> > Accuracy measures how close the basis is to the optimal solution. We follow the same equation as in [Fan et al., 2023] (page 13) to compute accuracy. In general, a closer basis tends to reduce the number of iterations, which can, in turn, shorten solving time. Therefore, achieving high accuracy is valuable. Additionally, it serves as an indicator of the GNN model’s ability to learn and approximate the optimal basis.
> >
> > However, a closer initial basis does not always guarantee greater acceleration. If the initial basis is invalid, Phase I may require significant time to obtain a valid one, which could still be far from optimal. While predicting a closer basis is an intuitive way to speed up the solver, additional refinements are necessary to achieve actual performance gains.
> >
> > Our work improves the model's ability to predict a closer basis. More importantly, we go beyond accuracy (closeness) by prioritizing actual solver acceleration. Through a detailed analysis of the LP problem, we introduce additional techniques that significantly enhance practical solving speed, which remains the ultimate objective.
>
>
> ### The tripartite graph should be tested against multiple GNN architectures (not just one) to demonstrate its effectiveness across different models.
> ### The impact of the tripartite representation should be better isolated from other factors (e.g., loss functions, preprocessing).
>
> > We add experiments on using tripartite graph-based GNN with difference bidirectional message passing layers (0, 2, 4 and 6 respectively) compared to the default solving mode without warm start. We use the base training without loss function for basic variable selection, the loss function for feasibility, and label preprocessing. Here are the number of iteration and solving time on the unpresolved datasets:
>
> > Iteration number:
> >
> | datasets       | n=0   | n=2   | n=4   | n=6   | default |
> | -------------- | ----- | ----- | ----- | ----- | ---- |
> | Mirp           | 18798 | 14017 | 12109 | 10813 | 25432 |
> | Anonymous      | 60146 | 23583 | 23166 | 21342 | 35330 |
> | Load_balancing | 4070  | 4357  | 4291  | 4512  | 7965 |
> | geomean        | 16633 | 11293 | 10637 | 10136 | 19271 |
>
> > Solving time in seconds:
> >
> | datasets       | n=0   | n=2   | n=4   | n=6   | default |
> | -------------- | ----- | ----- | ----- | ----- | ---- |
> | Mirp           | 8.29 | 7.43 | 7.07 | 6.39 | 11.17 |
> | Anonymous      | 7.19 | 7.12 | 6.75 | 6.87 | 9.08 |
> | Load_balancing | 16.71  | 10.49  | 10.98  | 8.85  | 7965 |
> | geomean        | 9.99 | 8.22 | 8.06 | 7.30 | 11.39 |
>
> > Due to time constraints, we have not yet completed testing on the Mirp2 dataset, but we will provide the results later. It is evident that bidirectional message passing is crucial as it effectively utilizes the coefficients in the constraint matrix. Additionally, adding more bidirectional message passing layers results in a slight improvement. We also plan to evaluate the performance using other GNN architectures, such as GraphTransformer and GAT, and will share the results once available.
>
> ### The paper correctly cites LP and MIP learning-based methods, but it would benefit from a broader discussion on alternative warm-starting techniques beyond GNNs.
>
> > Thanks for your suggestions. We will add more literature on warm-starting techniques and update the revision.

---

> > ### Comment · Reviewer_BL5r · 2025-04-02
> >
> > Thank you very much for the detailed and thoughtful response. The clarifications provided have addressed most of my concerns, especially regarding the distinction between basis accuracy and actual solver acceleration. I appreciate the effort to explain the nuanced relationship between closeness to the optimal basis and practical performance improvements.
> >
> > Regarding the point I raised earlier — “The tripartite graph should be tested against multiple GNN architectures (not just one) to demonstrate its effectiveness across different models. The impact of the tripartite representation should be better isolated from other factors (e.g., loss functions, preprocessing).” — I acknowledge that the authors have conducted ablation studies with different numbers of bidirectional message passing layers, which is helpful for understanding the influence of GNN depth. This is a valuable addition.
> >
> > However, my original concern was slightly broader. The current GNN architecture used in this paper appears to be based on the one proposed by Qasim et al. (2019). While this is a reasonable and relevant choice, the field of graph neural networks has evolved rapidly in recent years, with more expressive and powerful architectures such as Graph Attention Networks, Graph Transformers, and others being widely adopted in various domains.
> >
> > My question is: why was this particular architecture chosen over more recent alternatives? Was it based on empirical performance, computational considerations, or compatibility with the tripartite representation? I believe a brief discussion or justification of the architectural choice would make the contribution more robust and provide readers with better insight into the design decisions.
> >
> > Again, thank you for the substantial improvements and clarifications in the revision.

---

> > > ### Author Response · Authors · 2025-04-07
> > >
> > > We sincerely thank the reviewer for the detailed and thoughtful feedback. We are glad to hear that the clarifications regarding basis accuracy and solver acceleration were helpful.
> > >
> > > Regarding the choice of GNN architecture: we appreciate the reviewer’s suggestion to evaluate our tripartite representation across a broader range of architectures. As correctly noted, our current model is based on the architecture proposed by Qasim et al. (2019). We chose this architecture for three main reasons:
> > >
> > > ### Comparative Consistency:
> > > This architecture is also used in the state-of-the-art work by Fan et al., which our method builds upon. To ensure a fair and direct comparison, we adopted their GNN design as a baseline.
> > >
> > > ### Analytical Interpretability:
> > > We provide a detailed analysis of the message passing behavior of this architecture in Appendix A.2 (p.15). In particular, we aim for the amount of message passing to be proportional to the magnitude of the corresponding coefficients in the constraint matrix and the projection of changes in variables (or constraints) onto the respective rows (or columns). This behavior aligns well with the structure and nature of LP problems.
> > >
> > > ### Computational Efficiency:
> > > Compared to more recent models such as Graph Transformers, this architecture is lightweight and computationally efficient, making it more practical for real-world applications.
> > >
> > >
> > > To further address the reviewer’s concern, we have added results using Graph Transformer architectures on the unpresolved datasets (excluding mirp2 due to time constraints). The following tables summarize the results:
> > >
> > > Iteration count:
> > > | datasets       | bipartite-transformerConv   | tripartite-transformerConv   | tripartite-graphConv   |
> > > | -------------- | ----- | ----- | ----- |
> > > | Mirp           | 16022 | 16146 | 14017 |
> > > | Anonymous      | 4561 | 4804 | 4357 |
> > > | Load_balancing | 23946 | 21124  | 23583  |
> > > | geomean        | 12051 | 11790 | 11294 |
> > >
> > > Solving time in seconds:
> > > | datasets       | bipartite-transformerConv   | tripartite-transformerConv   | tripartite-graphConv   |
> > > | -------------- | ----- | ----- | ----- |
> > > | Mirp           | 7.24 | 6.96 | 6.04 |
> > > | Anonymous      | 9.87 | 9.75 | 5.74 |
> > > | Load_balancing | 6.37 | 6.44  | 10.61  |
> > > | geomean        | 7.69 | 7.59 | 7.17 |
> > >
> > > These results show a modest advantage of our tripartite representation over the bipartite baseline. Additionally, we observe that the GraphConv generally outperforms the TransformerConv, further supporting our claim that the GraphConv architecture aligns well with the structure of LP problems and is expressive enough to capture the necessary properties.
> > >
> > > While a full exploration of alternative GNN backbones is beyond the scope of this work, we view this as an important and promising direction for future research. We thank the reviewer again for raising this valuable point.

---

### Official Review · Reviewer_KgT2 · 2025-03-16

**Overall Recommendation:** 3

**Summary:**

The paper proposes  a new Graph Neural Network model for predicting the initial basis in the simplex method for solving linear programming (LP) problems. They use a tripartite graph that includes a global node and also nodes for dual variables, in addition to nodes for constraints and primal variables in a bipartite graph from previous work. Also, a new loss function is introduced to improve basic variable selection and basis feasibility. The proposed model significantly outperforms the state-of-the-art method in terms of prediction accuracy, reducing the number of iterations and solving time required by the LP solver. Effectiveness of each design component is evaluated through ablations study.

**Claims And Evidence:**

The paper presents experimental results on several standard LP datasets (Load Balancing, Anonymous, MIRP relaxed from MILP instances) to support their claims. They compare their model against the existing SOTA (bipartite GNNs) and the default solver setting. The tables and figures show improvements in prediction accuracy and solver performance metrics (iterations and time). They also include an ablation study to demonstrate the effectiveness of each component of their proposed method.

A few aspects could be improved:

* Comparison with other warm-start heuristics. The paper mentions a few non-ML methods for basis selection. It would be important to include them as a baseline, though previously probably has shown they are no better than the bipartite GNN.
* The instances used are from MILP benchmark with relaxed integral constraints. The paper doesn’t quite justify why these benchmarks are important for LP solving that translate to real-world impact. In other words, why not use benchmarks designed for LP instead of MILP?

**Essential References Not Discussed:**

Not that i'm aware of.

**Experimental Designs Or Analyses:**

The experiments are well-designed to demonstrated the effectiveness through prediction accuracies,  runtime and numbers of iterations.
However, at this point, it is unclear how effective the method is for larger/harder instances (e.g, instances that takes 5 minute or even more to solve).

**Methods And Evaluation Criteria:**

See above comments about benchmarks and other baselines.

In addition, the instances selected seem easy to solved, that is, on average the runtime is only less than a minute. It would be interesting to see how well this method performs on hard instances.

**Other Comments Or Suggestions:**

A writing issues: There should be a space between words and citations. Some of them are missing.

**Other Strengths And Weaknesses:**

No other strengths or weaknesses i want to point out.

**Questions For Authors:**

In ablation study, why is tripartite without P (label preprocessing) the best?
Are the improvements statistically significant?

**Relation To Broader Scientific Literature:**

The paper studies an important problem related to linear optimization.

**Theoretical Claims:**

N/A

---

> ### Author Rebuttal · Authors · 2025-04-01
>
> ### Comparison with other warm-start heuristics. The paper mentions a few non-ML methods for basis selection. It would be important to include them as a baseline, though previously probably has shown they are no better than the bipartite GNN.
>
> > Thanks for your suggestion! We will incorporate these non-ML methods as baselines in our paper.
>
> ### The instances used are from MILP benchmark with relaxed integral constraints. The paper doesn’t quite justify why these benchmarks are important for LP solving that translate to real-world impact. In other words, why not use benchmarks designed for LP instead of MILP?
>
> > The current pure LP benchmarks are very diverse instance by instance and large-sccale -- the reason may be partly due to the strong commercial solvers like Gurobi, CPlex and COPT. In fact, there is little benchmarks containing similar distributions for training and testing data to evluate learning-based methods, as the learning-based LP solvers are still in the early stage.
>
> > In contrast, the MILP benchmarks contain rich instances that can be divided into training and testing ones.
>
>
> ### In addition, the instances selected seem easy to solved, that is, on average the runtime is only less than a minute. It would be interesting to see how well this method performs on hard instances (e.g, instances that takes 5 minute or even more to solve).
>
> > Our approach scales effectively to large LP instances. In the Mirp2 dataset, some instances are so large that they require several minutes to several hours for the solver to process. We demonstrate the acceleration achieved on these large instances as follows:
> >
> | instances               | iter_default | iter_tripartiteBMP | time_default | time_tripartiteBMP |
> | ----------------------- | ------------ | ------------------ | ------------ | ------------------ |
> | LR1_DR04_VC05_V17b_t360 | 561299       | 409428             | 491.48       | 250.7              |
> | LR1_DR05_VC05_V25b_t360 | 706451       | 777452             | 474.74       | 513.63              |
> | LR1_DR08_VC05_V40b_t180 | 301789       | 309473             | 143.06       | 128.84              |
> | LR1_DR08_VC10_V40b_t120 | 267155       | 298875             | 112.76       | 131.6              |
> | LR1_DR12_VC10_V70a_t180 | 836561       | 619425             | 751.76       | 606.93              |
> | geomean | 484672       | 448942             | 309.27       | 265.74              |
> > The iter_* columns represent the number of iterations, while the time_* columns indicate the solving time in seconds. In practice, the GNN model is particularly useful for large LP problems, as its computational overhead is negligible compared to the overall solving time. However, for extremely large LPs—which translate to massive graphs—memory constraints may require the use of graph sampling techniques like GraphSAGE during both training and inference.
>
> ### A writing issues: There should be a space between words and citations. Some of them are missing.
>
> > Thanks and we have polished the paper.
>
> ### In ablation study, why is tripartite without P (label preprocessing) the best? Are the improvements statistically significant?
>
> > From Table 8, we observe that the only reason the tripartite model without P outperforms the one with P is due to a significant improvement on the Mirp1 dataset. This may be because, while processing the labels to reduce inconsistency helps the GNN learn and achieve better accuracy, a closer initial basis may sometimes be invalid, requiring more time in Phase I to repair it. However, this is not a general trend, as adding P consistently leads to better acceleration across all other datasets.

---

### Decision · Program_Chairs · 2025-05-01

**Decision:**

Accept (poster)

**Comment:**

The paper proposes a GNN to predict a basis for the simplex algorithm for solving LPs. Contributions consists in the graph model, loss function and strong experimental performance. Reviewers agree on the merits of the paper and recommend acceptance. The thoughtful rebuttal is positively acknowledged. We urge the authors to incorporate the minor criticism raised by reviewers and take the rebuttal discussion into account in a final version of the paper.